
# SciKit-GStat 1.0: A SciPy flavoured geostatistical variogram estimation toolbox written in Python.

Mirko Mälicke[1]

[1]Institute for Water and River Basin Management, Karlsruhe Institute of Technology (KIT), Germany
**Correspondence:** Mirko Mälicke (mirko.maelicke@kit.edu)

**Abstract.** Geostatistical methods are widely used in almost all geoscientific disciplines, i.e. for interpolation, re-scaling, data assimilation or modelling. At its core geostatistics aims to detect, quantify, describe, analyze and model spatial covariance of observations. The variogram, a tool to describe this spatial covariance in a formalized way, is at the heart of every such method. Unfortunately, many applications of geostatistics rather focus on the interpolation method or the result, than the quality of the estimated variogram. Not least because estimating a variogram is commonly left as a task for computers and some software implementations do not even show a variogram to the user. This is a miss, because the quality of the variogram largely determines, whether the application of geostatistics makes sense at all. Furthermore, the Python programming language was missing a mature, well-established and tested package for variogram estimation a couple of years ago.

Here I present SciKit-GStat, an open source Python package for variogram estimation, that fits well into established frameworks for scientific computing and puts the focus on the variogram before more sophisticated methods are about to be applied. SciKit-GStat is written in a mutable, object-oriented way that mimics the typical geostatistical analysis workflow. Its main strength is the ease of usage and interactivity and it is therefore usable with only a little or even no knowledge in Python. During the last few years, other libraries covering geostatistics for Python developed along with SciKit-GStat. Today, the most important ones can be interfaced by SciKit-GStat. Additionally, established data structures for scientific computing are reused internally, to keep the user from learning complex data models, just for using SciKit-GStat. Common data structures along with powerful interfaces enable the user to use SciKit-GStat along with other packages in established workflows, rather than forcing the user to stick to the authors programming paradigms.

SciKit-GStat ships with a large number of predefined procedures, algorithms and models, such as variogram estimators, theoretical spatial models or binning algorithms. Common approaches to estimate variograms are covered and can be used out of the box. At the same time, the base class is very flexible and can be adjusted to less common problems, as well. Last but not least, it was made sure, that a user is aided at implementing new procedures, or even extending the core functionality as much as possible, to extend SciKit-GStat to uncovered use-cases. With broad documentation, user guide, tutorials and good unit-test coverage, SciKit-GStat enables the user to focus on variogram estimation, rather than implementation details.





## 1 Introduction

Today, geo-scientific models are more available than they have ever been. Hence, producing in situ datasets to test and validate models is as important as ever. One challenge that most observations of our environment have in common is they are non-exhaustive and often only observe a fraction of the observation space. A prime example is the German national rainfall observation network. Considering the actual size of a Hellmann observation device the approx. 1900 stations, the meteorological service operates, sum up to only 38 m$^2$. Compared to the area of Germany, these are non-exhaustive measurements.

If one takes an aerial observation, such as a rainfall radar, into account, at face value this can seem to be different. But a rainfall radar is actually only observing a quite narrow band in height, which might well be a few thousand meters above ground (Marshall et al., 1947). And it observes the atmosphere's reflectivity, not the actual rainfall. Consequently, the *rainfall* input data for geo-scientific models, which is often considered to be an *observation*, is rather non-exhaustive or a product of yet another modelling or processing step. Methods that interpolate, merge or model datasets can often be considered to be geostatistical, or at least rely upon them (Goovaerts, 2000; Jewell and Gaussiat, 2015).

I hereby present SciKit-GStat, a Python package that implements the most fundamental processing and analysis step of geostatistics: the variogram estimation. It is open source, object oriented, well documented, flexible and powerful to overcome the limitation many current software implementation may have.

The successful journey of geostatistics started in the early 1950s and continuous progress has been made ever since. The earliest work was published 1951 by the South African engineer David Krige (Krige, 1951). He also lent his name to the most popular geostatistical interpolation technique *kriging*. Nevertheless, Matheron (1963) is often referenced as the founder of geostatistics. His work introduced the mathematical formalization of the variogram, which opened geostatistics to a wider audience, as it could easily be applied to other fields than Mining.

From this limited use case, geostatistics has gained importance and spread annually. A major review work is publish almost every decade, illustrating the continuous progress of the subject. Today, it's a widely accepted field that is used throughout all disciplines in geoscience. Dowd (1991) reviewed the state of the art works from 1987 to 1991 in the fields of geostatistical simulation, indicator kriging, fuzzy kriging and interval estimation. But also more specific applications such as hydrocarbon reservoirs and hydrology are reviewed. Atkinson and Tate (2000) reviewed geostatistical works specifically focused on scale issues. The authors highlight the main issues and pitfalls when geostatistics are used to upscale or downscale data, especially in remote sensing and GIS. A few years later, Hu and Chugunova (2008) summarized 50 years of progress in geostatistics and compared it to more recent developments in multi-point geostatistics. These methods infer needed multivariate distributions from the data to model covariances. Recently, Ly et al. (2015) reviewed approaches for spatial interpolation, including geostatistics. This work focuses on the specific application of rainfall interpolation needed for hydrological modelling. Such works are only a small extract from what has been published during recent years. They are only outnumbered by the many domain-specific studies that focus on improving geostatistical methods for specific applications.

In recent years the field of geostatistics has experienced many extensions. Many processes and their spatial patterns studied in geoscience are not static but dynamically change on different scales. A prime example is soil moisture, which changes





on multiple temporal scales exposing spatial patterns that are not necessarily driven by the same processes throughout the

year (Western et al., 2004; Vereecken et al., 2008; Vanderlinden et al., 2012; Mälicke et al., 2020). The classic Matheronian geostatistics assumes stationarity for the input data. Hence, a temporal perspective was introduced into the variogram, modeling the spatial covariance accompanied by its temporal counterpart (Christakos, 2000; Ma, 2002, 2005; De Cesare et al., 2002). In parallel, approaches were developed, that questioned and extended the use of Euclidean distances to describe proximity between observation locations (Curriero, 2005; Boisvert et al., 2009; Boisvert and Deutsch, 2011). Last but not least, efforts

are made to overcome the fundamental assumption of Gaussian dependence, that underlies the variogram function. This can be achieved for example by sub-Gaussian models (Guadagnini et al., 2018) or copulas (Bárdossy, 2006; Bárdossy and Li, 2008). Non-Gaussian geostatistics are, however, not covered in SciKit-GStat.

The variogram is the most fundamental means of geostatistics and a prerequisite to apply other methods, such as interpolation. It relates the similarity of observations to their separating distance using a spatial model function. This function, bearing

information about the spatial covariance in the dataset, is used to derive weights for interpolating at unobserved locations. Thus, any uncertainty or error made during variogram estimation, will be propagated into the final result. As described, geoscientific datasets are often sparse in space and that makes it especially complex to choose the correct estimator for similarity and decide when two points are considered *close* in space. Minor changes to spatial binning and aggregations can have a huge impact on the final result, as will be shown in this work. This is an important step that should not entirely be left

to the computer. To foster the understanding and estimation of the variogram, SciKit-GStat is equipped with many different semi-variance estimators (table 1) and spatial models (table 2), where other implementations only have one or two options if any at all. Spatial binning, can be carried out utilizing one of ten different algorithms to break up the tight corset that geostatistics usually employs for this crucial step. Finally, SciKit-GStat implements various fitting procedures, each one in weighted and unweighted variation, with many options to automate the calculation of fitting weights. These tools enable a flexible and

intuitive variogram estimation. Only then, is the user able to make an informed decision, whether a geostatistical approach is even the correct procedure for a given dataset at all. Otherwise, Kriging would interpolate based on a spatial correlation model, which is in reality not backed-up with data.

De-facto standard libraries for geostatistics can be found in a number of commonly used programming languages. In FOR-TRAN, there is `gslib` (Deutsch and Journel, 1998), a comprehensive toolbox for geostatistical analysis and interpolation.

Spatio-temporal extensions to gslib are also available (De Cesare et al., 2002). For the R programming language, the `gstat` package (Pebesma, 2004; Gräler et al., 2016) can be considered the most complete package, covering most fields of applied geostatistics.

For the Python programming language, there was no package comparable to `gstat` in 2016. A multitude of Python packages, that were related to geostatistics could be found. A popular geostatistics related Python package is `pykrige` (Murphy et al.,

2021). As the name already implies, it is mainly intended for kriging interpolation. The most popular kriging procedures are implemented, however, only limited variogram analysis is possible. HPGL is an alternative package offering very comparable functionality. Unlike pykrige, the library is written in C++, which in wrapped and operated through Python. The authors claim





to provide a substantially faster implementation than gslib (which is written in FORTRAN). Another geostatistical Python library that can be found is *pygeostat*. It mainly focuses on geostatistical modeling. Unfortunately, obtaining the files and then
installing it in a clean Python environment turned out to be cumbersome[1].

All of the reviewed packages focus only on a specific part of geostatistics and in general, interfacing options were missing. Thus, I decided to develop an open source geostatistics package for the Python programming language called **SciKit-GStat**. In the course of the following years, another Python package with similar objectives was developed called `gstools` (Müller and Schüler, 2021). Both packages emerged at similar times; SciKit-Gstat was first published on Github in July 2017, `gstools` in
January 2018. With streamlining developments between these two packages, the objective of SciKit-GStat shifted and is today mainly focused on variogram estimation. Today, both packages work very well together and the developers of both packages collaborate to discuss and streamline future developments. Further details driving this decision are stated throughout this work, especially in section 2.2, 4.2 and 5.3. One of the goals of this work is to present differences between SciKit-GStat and existing other packages and illustrate, how it can be interfaced and connected to them. This will foster the development of a unique
geostatistical working environment that can satisfy any requirement in Python.

A number of works were especially influential during the development of SciKit-GStat. An early work by Burgess and Webster (1980) published a clear language description, of what a variogram is and how it can be utilized to interpolate soil properties to unknown locations. In the same year Cressie and Hawkins (1980) published an alternative variogram estimator to the Matheron estimator introduced 20 years earlier. This estimator is an important development, as its contained power trans-
formation makes it more robust to outliers, that we often face in geoscience. A noticeable amount of functions implemented in SciKit-GStat are directly based on equations provided in Bárdossy and Lehmann (1998). This work does not only provide a lot of statistical background to the applied methods, but also compares different approaches for kriging. Finally, a practical guide to implement geostatistical applications was published by Montero et al. (2015). A number of model equations implemented in SciKit-GStat are directly taken from this publication.

SciKit-GStat is a toolbox that fits well into the scipy environment. For scientific computing in Python, SciPy (Jones et al., 2001–) has developed to be the de facto standard environment. Hence, using available data structures, such as the numpy array (van der Walt et al., 2011), as an input and output format for SciKit-GStat functions makes it very easy to integrate the package into existing environments and workflows. Additionally, SciKit-GStat uses SciPy implementations for mathematical algorithms or procedures wherever available and feasible. I.e., the SciPy least squares implementation is used to fit a variogram
model to observed data. Using this common and well-tested implementation of least squares makes SciKit-GStat less error prone and fosters comparability to other scientific solutions also based on SciPy functionality.

SciKit-GStat enables the user to estimate standard, but also more exotic variograms. This process is aided by a multitude of helpful plotting functions and statistical output. In other geostatistical software solutions, the estimation of a variogram is
often left entirely to the computer. Some kind of evaluation criterion or objective function takes the responsibility of assessing

---

[1]At the time, several undocumented issues raised and solving them was not straightforward



the variograms suitability for expressing the spatial structure of the given input data in a model function. Once used in other geostatistical applications, such as kriging, the theoretical model does not bear any information about its suitability or even goodness of fit to the actual experimental data used. Further advanced geostatistical applications do present a variogram to the user, while performing other geostatistical tasks, but this often seems as a passive information that the user may recognize

or ignore. The focus is on the application itself. This can be fatal as the variogram might actually not represent the statistical properties well enough. One must remember that the variogram is the foundation of any geostatistical method and unnoticed errors within the variogram will have an impact on the results even if the maps look viable. The variogram itself is a crucial tool for the educated user to interpret whether data interpolation using geostatistics is valid at all.

SciKit-GStat takes a fundamentally different approach here. The variogram itself is the main result. The user may use a

variogram and pass it to a kriging algorithm, or use one of the interfaces to other libraries. However, SciKit-GStat makes this manual step by design. The user is put from a passive into an active role, and is therefore, close to geostatistical textbooks, where the variogram is always the first geostatistical method introduced.

SciKit-GStat is also designed for educational applications. Both students and instructors are specifically targeted within SciKit-GStat's documentation and user guide. While some limited knowledge of the Python programming language is assumed,

the user guide starts from zero in terms of geostatistics. Beside a technical description of the SciKit-GStat classes, the user is guided through the implementation of the most important functionality. This fosters a deeper understanding of the underlying methodology for the user. By using SciKit-GStat documentation, a novice user does not only learn how to use the code, but also what it does. This should be considered a crucial feature for scientific applications, especially in geostatistics where a multitude of one-click software is available, producing questionable results if used by uneducated users.

SciKit-GStat is well documented and tested. The current unit test coverage is >90%. The online documentation includes an installation guide, the code reference and an user guide. Additionally, tutorials are available, that are suitable for use in higher education level lectures. To facilitate an easy usage of the tutorials a Docker image is available (and the Dockerfile is part of SciKit-GStat). SciKit-GStat has a growing developer community on Github and is available under a MIT license.

The following section will give a more detailed overview of SciKit-GStat. Section 3 introduces the fundamental theory

behind geostatistics as covered by SciKit-GStat. Section 4 guides through the specific implementation of the theory, section 5 gives details on user support and contribution guidelines.

## 2   SciKit-GStat general overview

The source code repository contains the Python package itself, the documentation and sample data. This work will focus mainly on the Python package, starting with a detailed overview in section 2.2. The documentation is introduced to some detail

in section 2.2 and section 5. Most data distributed with the source code is either artificially created for a specific chapter in the documentation, or originally published somewhere else. In these cases either the reference or license is distributed along with



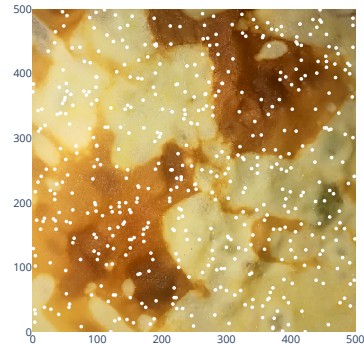

**Figure 1.** Original photograph of the pancake used to generate the pancake dataset. The white points indicate the 500 sampling locations that were chosen randomly, without repeating. The observation value is the red channel value of the RGB value of the specified pixel.

the data itself. For this publication, all figures were created with the same data, wherever suitable. This is further introduced in section 2.1 and appendix A.

## 2.1 Data

There are already some benchmark datasets for geostatistics, such as the meuse dataset distributed with the R package `gstat` (Pebesma, 2004), which is also included in SciKit-GStat. In order to provide a dataset of a random field (not only a sample thereof), which has obvious spatial covariance structure, an image of a pancake was utilised (figure 1). This approach was employed to enable the implementation of custom sampling strategies and the ability to analyse the dataset at any level of sampling density within such an image. Furthermore, with a pancake, one does not focus too much on location specifics or

properties of the random field, as it will happen with i.e. a remote sensing soil moisture product from an actual location on earth. The pancake browning (figure 1) shows a clear spatial correlation, the field is exhaustive at the resolution of the camera device and creating new realizations of the field is possible as well. Processes forming spatial structure in browning might be different from processes dictating the spatial structure of i.e. soil moisture, but they are ultimately also driven by physical principles. Testing SciKit-GStat tools not only with classic geoscientific data, but also with pancakes made the implementation

more robust. But it also illustrates that the geostatistical approach holds beyond geoscience. A technical description of how to cook your own dataset is given in appendix A.

## 2.2 Package description

SciKit-GStat is a library for geostatistical analysis written in the Python programming language. The Python interpreter must be of version 3.6 or later. The source files can be downloaded and installed from the Python package index using pip, which is the



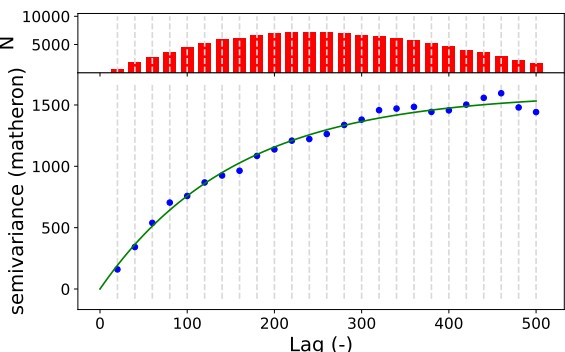

**Figure 2.** Default variogram plot of SciKit-GStat using the matplotlib backend. The variogram was estimated with the pancake dataset using the exponential model fitted to a experimental variogram resolved to 25 evenly spaced lag classes, up to 500 units (the axe length of the sampled field).

standard tool for Python 3[2]. All dependencies are installed along with the source files. This is the standard and recommended procedure for installing and updating modules in Python 3. Additionally, the source code is open and available on Github and can be downloaded and installed from source. SciKit-GStat is published under a MIT license.

The presented module is built upon common third party packages for scientific computing in Python, called `scipy`. In recent years, the SciPy ecosystem has become de facto standard for scientific computing and applications in Python. SciKit-GStat

makes extensive use especially of `numpy` (Oliphant, 2006; van der Walt et al., 2011) to build data structures and numerical computations, `matplotlib` (Hunter, 2007) and `plotly` (Inc., 2015) for plotting and the `scipy` library itself (Jones et al., 2001–) for solving some specific mathematical problems, such as least squares or matrix operations.

An object oriented programming approach was chosen for the entire library. SciKit-GStat is designed to interact with the user through a set of classes. Each step in a geostatistical analysis workflow is represented by a class and its methods. Argument

names passed to an instance on creation are chosen to be as close as possible to existing and common parameter names from geostatistical literature. The aim is to make the usage of SciKit-GStat as intuitive as possible for geoscientists with only little or no experience in Python.

The main focus of the package is variogram analysis. Ordinary kriging is also implemented into SciKit-GStat, but the main strength is variogram analysis. Kriging is available as a valuable tool to cross-validate the variogram by interpolating the

observation values. For flexible, feature rich and fast kriging applications, the variogram can be exported to other libraries with ease. SciKit-GStat offers an extensible and flexible class that implements common settings out of the box but can be adjusted to rather uncommon problems with ease. An example variogram is shown in figure 2. By default, the user has an experimental

---

[2]SciKit-GStat is also available on Conda Forge, the largest community driven Anaconda channel. This package is not covered here, as the content is the same and installation requires the presence of an Anaconda environment and some knowledge of the system. Nevertheless, Anaconda is widely spread among scientists and it might be worth mentioning the existence for anaconda users.





**Table 1.** Overview of all semi-variance estimator functions implemented in SciKit-GStat. Using *Normalized Range* and *Percentile* is only advised to users understanding the implications as explained in section 4.1.3.

| Estimator | Identifier | Description | Reference |
|---|---|---|---|
| Mathéron | `'matheron'` | Default, most popular estimator | Matheron (1963) |
| Cressie-Hawkins | `'cressie'` | Power transformation based - robust to outliers | Cressie and Hawkins (1980) |
| Dowd | `'dowd'` | Median based, fast estimator for non-normal distributed residuals | Dowd (1984) |
| Genton | `'genton'` | Percentile-based estimator - powerful for skewed residuals, but very computationally intensive | Genton (1998) |
| Shannon Entropy | `'entropy'` | Information theoretic measure focusing information content of residuals | Shannon (1948) |
| Normalized Range | `'minmax'` | Experimental estimator using only the spread of residuals | |
| Percentile | `'percentile'` | Uses any user-defined percentile as semi-variance, but untransformed. Experimental | |

variogram, a well fitted theoretical model and a histogram to estimate the point pair distribution in the lag classes at ones disposal. This way, the plot of the variogram instance helps the user at first sight to not only estimate goodness of fit, but also
the spatial representativity of the variogram for the sample used. All parameters can be changed in place and the plot can be updated, without restarting Python or creating new unnecessary variables and instances.

SciKit-GStat contains eight different semi-variance estimators (overview in table 1) and seven different theoretical variogram model functions (overview in table 2). At the same time implementing custom models and estimators is supported by a decorator function that only requires the mathematical calculation from the user, which can be formulated with almost no prior
Python knowledge. Often with a single line of code.

SciKit-GStat offers a multitude of customization options to fit variogram models to experimental data. The model parameters can be fitted manually or by one of three available optimization algorithms: Levenberg-Marquardt, Trust Region Reflective and Maximum Likelihood (see section 4.1.5). It is also possible to combine both. Furthermore, it is possible to weight experimental data. Such weighting of experimental data is a crucial feature to make a variogram model fit data at short lags more precisely
than distant observations. The user can manually adjust weights or use one of the many predefined functions, that define weights i.e. dependent on the separating distance. Closely related is the way how SciKit-GStat handles spatial aggregation. The user can specify a function that will be used to calculate an empirical distribution of separating distance classes, which are the foundation for spatial aggregation. Especially for sparse datasets which base their aggregation on small sample sizes, even adding or removing a single lag class can dramatically change the experimental variogram. The default function defines
equidistant distance lag classes, as mostly used in literature. However, SciKit-GStat also includes functionality for auto-deriving


**Table 2.** Overview of all theoretical variogram model functions implemented in SciKit-GStat.

| Model | Identifier | Description | Implementation |
|---|---|---|---|
| Spherical | 'spherical' | Short ranged correlation length, popular model in geoscience; for smooth, but steep gradients in fields. | Burgess and Webster (1980) |
| Exponential | 'exponential' | Long ranged for smooth fields with less steep gradients. | Journel and Huijbregts (1976) |
| Gaussian | 'gaussian' | Mid ranged for sharply changing fields | Journel and Huijbregts (1976) |
| Cubic | 'cubic' | Similar to Gaussian models, but with a shorter correlation length. | Montero et al. (2015) |
| Matérn | 'matern' | Has an additional smoothness parameter to adapt shapes between Exponential and Gaussian models. | Zimmermann et al. (2008) |
| Stable | 'stable' | Has an additional shape (power) parameter to adapt the range. | Montero et al. (2015) |
| Isotonic Regression | 'harmonize' | Data harmonization algorithm to directly monotonize the experimental variogram, without fitting | Pedregosa et al. (2011) |

a suitable number of lag classes or cluster based methods, which have to my knowledge, not been used so far in this context. A complete overview all all functions is given in table 3.

Interfaces to a number of other geostatistical packages are provided. SciKit-GStat defines either an export method or a conversion function to transform objects that can be read by other packages. Namely, the Variogram can export an parameterized
custom variogram function, which can be read by Kriging classes of the `pykrige` package. A similar export function can transform a variogram to a covariance model as used by `gstools`. This package is evolving to be the prime geostatistical toolbox in Python. Thus, a powerful interface is of crucial importance. Finally, a wrapping class for Variogram is provided that will make it accessible as a scikit-learn (Pedregosa et al., 2011) estimator object. This way, scikit-learn can be used to perform parameter search and use variograms in a machine learning context.

SciKit-GStat is easily extensible. Many parts of SciKit-GStat were designed to keep the main algorithmic functions clean. Overhead, like type checks and function mapping to arrays are outsourced to instance methods wherever possible. This enables the user to implement custom functions with ease, even if they are not too familiar with Python. As an example, implementing a new theoretical model is narrowed down to only implementing the mathematical formula this way.

Documentation provided with SciKit-Gstat are tailored for educational use. The documentation mainly contains a user
guide, tutorials and a technical reference. The user guide for SciKit-GStat does not have any prerequisites in geostatistics and guides the reader through the underlying theory, while walking through the implementation. For users with some experience in Python, geostatistics and other fields of statistics, tutorials are provided. The tutorials focus on a specific aspect of SciKit-GStat and demonstrate the application of the package. Here, a sound understanding of geostatistics is assumed. Finally, the





**Table 3.** Overview of all lag class binning methods implemented in SciKit-GStat.

| Function | Identifier | Description | Implementation |
|---|---|---|---|
| Equidistant lags | 'even' | $N$ lags of same width; Almost always used. | Mälicke et al. (2021) |
| Uniform lags | 'uniform' | $N$ lags of same sample size; Estimats are based on the same sample size & no empty bins | Mälicke et al. (2021) |
| Sturge's rule | 'sturges' | Equidistant lags derived from Sturge's rule; use for small normal distributed distance matrices | Jones et al. (2001–) |
| Scott's rule | 'scott | Equidistant lags derived from Scott's rule; use for large datasets | Jones et al. (2001–) |
| Freedman-Diaconis estimator | 'fd' | Equidistant lags; use for small datasets with outliers in the distance matrix | Jones et al. (2001–) |
| Square-root | 'sqrt' | Equidistant lags; Very fast function, but usually not recommended | Jones et al. (2001–) |
| Doane's rule | 'doane' | Equidistant lags; based on data skewness, use for small non-normal distance matrices | Jones et al. (2001–) |
| K-Means | 'kmeans' | Non-equidistant lags; clustered distance matrix is used as binning; slow but statistically robust | Pedregosa et al. (2011) |
| Hierachical Clusters | 'ward' | Non-equidistant lags; clustered distance matrix is used as binning; Based on Ward's criterion for minimizing cluster variance. Computational intensive | Pedregosa et al. (2011) |
| Stable Entropy | 'stable_entropy' | Non-equidistant lags; Bin edges are set by minimizing the deviations of per-lag Shannon entropy | Mälicke et al. (2021) |

technical reference does only document the implemented functions and classes from a technical point of view. It is mainly

designed for experienced users that need an in-depth understanding of the implementation or for contributors that want to extend SciKit-GStat.

SciKit-GStat is 100% reproducible through docker images. With only the docker software installed (or any other software that can run docker containers), it is possible to run the scikit-gstat docker image, which includes all dependencies and common development tools used in scientific programming. This makes it possible to follow the documentation and tutorials instantly.

The user can use a specific SciKit-GStat version (from 1.0 on) and conduct analysis within the container. That will fix all used software versions and, if saved, make the analysis 100% reproducible. At the same time the installation inside docker container does not affect any existing Python environment on the host system and is therefore perfect to test SciKit-GStat.

SciKit-GStat is recognized on Github and has a considerable community. Issues and help requests are submitted frequently and are usually answered in a short amount of time by the author. At the same time, efforts are made to establish a broader

developer community, to foster support and development. Additionally, the development on SciKit-GStat is closely coordinated with `gstools` and the parenting Geostat-Framework developer community.





## 3 Main geostatistical components

### 3.1 Variogram

In geostatistical literature, the terms *semi-variogram* and *variogram* are often mixed or interchanged. Although closely related,
two different methods are described by these terms. In most cases, the *semi-variogram* is used, but called simply *variogram*.
Here, I follow this common nomenclature and both terms describe the semi-variogram in the following.

At its core, the semi-variogram is a means to express how spatial dependence in observations changes with separating distance. An observation is here defined to be a sample of a spatial random function. While these functions are usually two or three dimensional in geostatistical applications, they can be N-dimensional in SciKit-GStat (including 1D). A more compre-
hensive and detailed introduction to random functions in the context of geostatistics is given in Montero et al. (2015, chapter 2.2, p. 11 ff.). The most fundamental assumption that underlies a variogram, is therefore, that proximity in space leads to similar observations (proximity in value). To calculate spatially aggregated statistics on the sample, the variogram must make an assumption up to which distance two observations are still close in space. This is carried out by using a distance lag over the exact distances, as two point pairs will hardly be at exactly the same spatial distance in real world datasets.

Separating distance is calculated for observation point pairs. For different distance lag classes (e.g. 10 m to 20 m), all point pairs $s_i, s_j$ within this class are aggregated to one value of (dis)similarity, called semi-variance $\gamma$. A multitude of different estimators are defined to calculate the semi-variance. For a specific lag distance $h$ (e.g. 10 m), the most commonly used Matheron estimator (Matheron, 1963) is defined by equation (1):

$$\gamma(h) = \frac{1}{2N(h)} \sum_{i=1}^{N(h)} (Z(s_i) - Z(s_{i+h})) \tag{1}$$

Where $N(h)$ is the number of point pairs for the lag $h$ and $Z(s)$ the observed value at the respective location $s$. The obtained functions is called an *experimental variogram* in SciKit-GStat. In literature, the term *empirical variogram* is also quite often used and is referring more or less, to the same thing. In SciKit-GStat, the empirical variogram is the combination of the lag classes and the experimental variogram. All estimators implemented in SciKit-GStat are described in detail in section 4.1.3.

To model spatial dependencies in a data set, a formalized mathematical model has to be fitted to the experimental variogram.
This step is necessary, to obtain parameters from the model, in a formalized manner. These describe spatial statistical properties of the model, which may (hopefully) be generalizable to the data set. These parameters are called variogram parameters and include:

1. *nugget* - the semi-variance at lag $h = 0$. This is the variance, that cannot be explained by a spatial model and is inherit to the observation context. (i.e. measurement uncertainties or small-scale variability).

2. *sill* - the upper limit for a spatial model function. The nugget and sill add up to the sample variance.

3. *effective range* - the distance, at which the model reaches 95% of the sill. For distances larger than the range, the observations become statistically independent. Variogram model equations also define a **model** parameter called range,



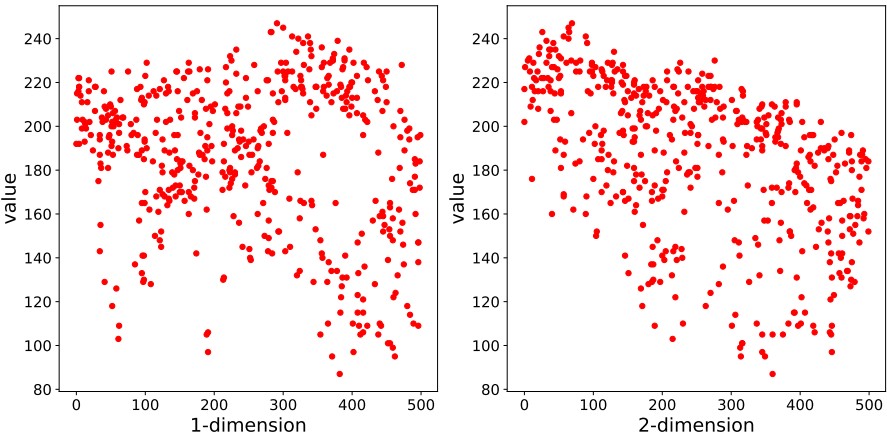

**Figure 3.** Location Trend plot as created by SciKit-GStat on a 2D dataset. The coordinates are disassembled into each provided dimension and the observation values are plotted relation. The user can quickly identify trends.

which leads to misunderstandings in the geostatistical community. To overcome these problems, SciKit-GStat formulated all implemented models based on the *effective range* of the variogram and not the range model parameter. Consequently, the given formulas might differ from some common sources by the transformation of effective range to range model parameter. This transformations are straightforward and reported in literature, but for some models (i.e. Gaussian) not commonly the same. In these cases, the user is encouraged to carefully check the implementation used in SciKit-GStat.

Closely related to these parameters is the nugget to sill ratio. It is interpreted as the share of spatially explainable variance in the sample and is therefore a very important metric to reject the usage of a specific variogram model at all.

The theoretical model is a prerequisite for spatial interpolation. For this to happen, a number of geostatistical assumptions need to be fulfilled. Namely, the observations have to be of second-order stationarity and the intrinsic hypothesis has to hold. This can be summarized as the requirement, that the expected value of the random function and its residuals must not dependent on the location of observation, but solely on the distance to other points. This assumption has to hold for the full observation space. Hence, the semi-variance is calculated with the distance lag $h$ as the only input parameter. A more detailed descriptions of these requirements is e.g. given in Montero et al. (2015, chapter 3.4.1 p. 27 ff.) or (Burgess and Webster, 1980; Bárdossy and Lehmann, 1998). An important tool to learn about trends in the input dataset is a scatter plot like shown in figure 3. The same variogram instance that was used for figure 2 is used here. Along the x-axis (1.dimension), there is no trend visible, but along the y-axis high valued observations seem to drop with increasing coordinate location. This readily available plot is useful to guide the user into the decision of utilizing statistical trend tests to test for statistical significance and finally detrending input data.

The other requirement for variogram models is that it has to be monotonically increasing. A drop in semi-variance would imply that observations become more similar with increasing distance, which is incompatible to the most fundamental as-



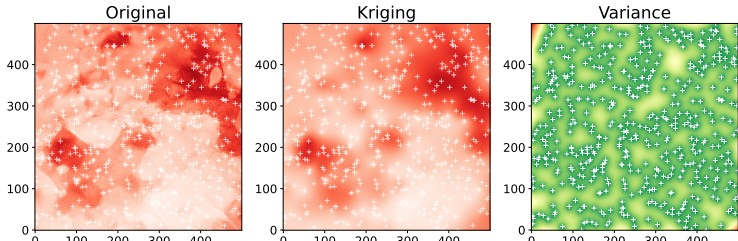

**Figure 4.** Ordinary Kriging result of the pancake dataset sample used in figure 2 and 3. The Kriging was performed with default parameters on a grid of same resolution as the original field. The white crosses indicate the sample positions.

sumption in geostatistics of spatial proximity. This requirement can only be met by a statistical model function and not the experimental variogram, which is often not monotonically increasing in a strict sense. This may happen due to the fact that
(spatial) observations are not exhaustive and measurements might be uncertain.

### 3.2 Kriging

One of the most commonly used applications of geostatistics is kriging. A sample result is shown in figure 4. The interpolation was made with the same variogram instance used to produce figures 2 and 3. The center sub-figure shows the result itself, along with the original field (left) and a kriging error map (right), which will be introduced later. In this example, the spatial
properties and correlation lengths of the original are well captured by the result.

Kriging estimates the value for an unobserved location $s_0$ as the weighted sum of nearby observations as shown in equation (2).

$$Z^*(s_0) = \sum_{i=1}^{N} \lambda_i Z(s_i) \tag{2}$$

Where $Z^*(s_0)$ is the estimation and $\lambda_i$ are the weights for the $N$ neighbors $s_i$. The kriging procedure uses the theoretical
variogram model fitted to the data to derive the weights from the spatial covariance structure. Furthermore, by requiring all weights to sum up to one (equation (3)) the unbiasedness of the prediction is assured.

$$\sum_{i=1}^{N} \lambda_i = 1 \tag{3}$$

A single weight can thereby be larger than one or smaller than 0. As the weights are inferred from the spatial configuration of the neighbors, this can require stronger influence ($\lambda > 1$) or even negative influence ($\lambda < 0$) of specific observations. Combined
with unbiasedness, this is one of the most important features of a kriging interpolation and can make it superior to, i.e. spline-based procedures in an environmental context. Deriving weights from the spatial properties of the data is especially helpful, as the local extreme values have likely not been observed, but their influence is present in the spatial covariance of the field close to it.

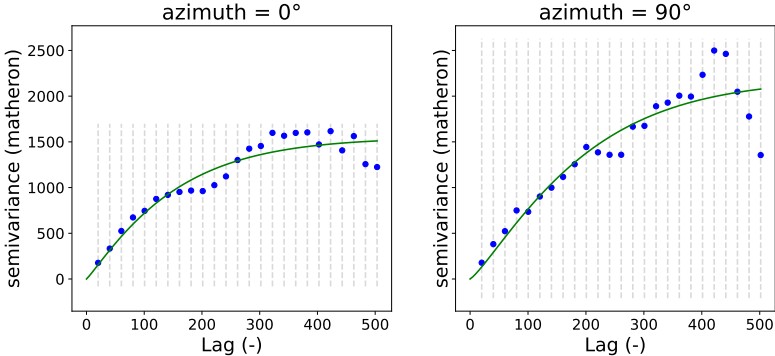

**Figure 5.** Two directional variograms calculated for the pancake dataset. Both variograms use the same parameters as the instance used to produce figure 2. In addition, the direction is taken into account. The two variograms shown differ only in the azimuth used, which is $0°$ (left) and $90°$ (right).

To obtain the weights for one unobserved location, a system of equations is formulated, called the kriging equation system (KES). By expecting the prediction errors to be zero (equation (4)) and substituting equation (2) in equation (4), the KES can be formulated.

$$E[^*(s_0) - Z(s_0)] = 0 \qquad (4)$$

The final kriging equation (5) is taken from Montero et al. (2015, equation 4.16, p. 86) and its derivation is given in chapter 4.3.1 of the same source (Montero et al., 2015, p.84-90).

$$\begin{cases} \sum_{j=1}^{N} \lambda_j \gamma(s_i - s_j) + \alpha = \gamma(s_i - s_0), & i = 1, \ldots, N \\ \sum_{i=1}^{N} \lambda_i = 1 \end{cases} \qquad (5)$$

Where $\alpha$ is the Lagrange multiplier needed to solve the KES by minimizing the estimation variance subject to the constraint of equation (3). By minimizing the prediction variance and requiring the weights to sum to one, it is possible to obtain the best linear, unbiased estimation. Thus, kriging is often referred to as being a BLUE (Best Linear Unbiased Estimator). Using Kriging an estimate of the variance of the spatial prediction can be obtained. This is shown in the right panel of figure 4. Such information is vital to assess the quality of the prediction. Finally, the setup of Kriging makes it a smooth interpolation, as the predictions very close to observation locations are approaching the observation values smoothly. The kriging variance is significantly higher in less densely sampled regions (figure 4), which enables the user to visually assess the spatial representativity of the obtained results.





### 3.3 Directional variogram

The standard variogram as described in section 3.1 handles *isotropic* samples. That means the spatial correlation length of the random field is assumed to be of comparable length in each direction. Usually, one refers only to the directions along the main coordinate axes. However, direction can be defined with any azimuth angle and does not have to match the coordinate axes. If the spatial correlation length differs in direction this is referred to as *anisotropy*. There are two different kinds of anisotropy: geometric and zonal anisotropy (Wackernagel, 1998). Considering geometric anisotropy, the effective range differs for the two

perpendicular main directions of the anisotropy. In the zonal case, sill and range differ. Geometric anisotropy can be handled by a coordinate transformation (Wackernagel, 1998). These cases can be detected by directional variograms. For an application, the main directions of anisotropy must be identified to then estimate an isolated variogram for each direction.

For each directional variogram, only point pairs are considered that are oriented in the direction of the variogram. For two observation locations $s_1, s_2$ the orientation is defined as the angle between the vector $u$ connecting $s_1$ and $s_2$ and a vector along

the first dimension axis: $e = [1, 0]$. The cosine of the orientation angle $\Theta$ can be calculated using equation (6):

$$cos(\Theta) = \frac{u \circ e}{|e| \cdot |(1,0)|} \tag{6}$$

The directional variogram finally defines an azimuth angle, defined analogous to equation (6) and a tolerance. Any point pair which deviates less than tolerance from the azimuth, is considered to be oriented in the direction of the variogram and will be used for estimation.

The example data used so far shows a small anisotropy (figure 5). The two variograms used exactly the same data and parameters as used for figure 2. The only difference is that both are directional and they use two different directions of $0°$ and $90°$. There is a difference in effective range and sill in the $90°$ directional variogram.

As long as more than one directional variogram is estimated for a data sample, the difference of the estimated variogram parameters describes the degree of anisotropy. In a kriging application, the data sample can now be transformed along the main

directions at which the directional variograms differ until the directional variograms do not indicate an anisotropy anymore. The common variogram of the transformed data can be used for Kriging and the interpolated field is finally transformed back. Transformations are not part of SciKit-GStat. The `scipy` and `numpy` packages offer many approaches to apply transformations. Alternatively, `gstools` implements anisotropy directly and can use it for covariance models and kriging. In these cases the user needs to identify the directions manually and specify them on object creation.

### 3.4 Space-time variogram


At the turn of the millennium, geostatistics had emerged to a major tool in environmental- and geoscience, the demand for new methods was rising. Datasets collected in nature are usually dynamic in time, which can easily violate the second order stationarity assumptions underlying classic geostatistics. Hence, substantial progress had been made to incorporate temporal dimensions into variograms.

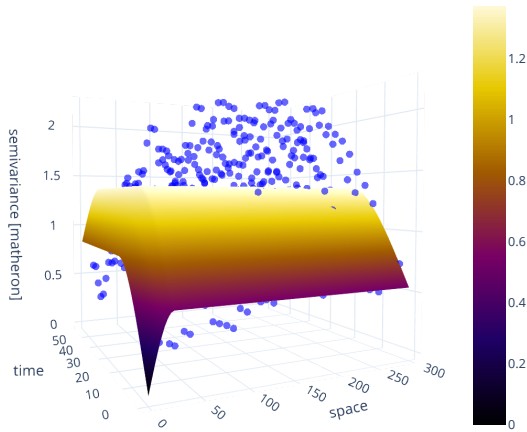

**Figure 6.** Default 3D scatter plot of a space-time variogram (blue points), with fitted product-sum model (surface). The variogram is estimated from the in-situ soil temperature measurements at 20cm depth (WSN product) published in Fersch et al. (2020). To decrease the computational workload, only every sixth measurement was taken from the timeseries.

The classic variogram is modelling the semi-variance of a sample in dependence of the separating distance of the underlying point pairs. For a space-time variogram, this dependence is expanded to time-lags. That means the data is not only segmented in terms of spatial proximity, but also temporal proximity. The resulting model will be capable to identify co-variances over space and time at the same time (figure 6). SciKit-GStat uses a 3D plot by default. The plot can be customized and exclude the fitted model or plot the experimental variogram rather as a surface, than a scatter plot. While figure 6 might contain both, the

experimental and the theoretical variogram, it is also quite overloaded and not always helpful. Finally, a printed 3D plot cannot be rotated, and the usage in publications is discouraged. To overcome these limitations, SciKit-GStat implements 2D contour plots of the experimental variogram in two variations, which differ only in visualization details (figure 7). The contour plot is the more appropriate means to inspect the covariance field as estimated by the space-time variogram. With the given example, one can see that the auto-correlation (temporal axis) is dominant and except for a few temporal lags (50 - 60, or 30 - 40),

the variogram shows almost a pure nugget along the spatial axis. Note that the contour lines smooth out the underlying field to close lines to rings wherever possible. This can lead to the impression that the experimental variogram is homogeneously smooth along the two axis. In fact, this is not the case and the smoothing is due to the implementation of contour lines. Thus, the contour plot should be used to get a general idea of the experimental variogram. To inspect the actual semi-variance values, the experimental variogram can be accessed and plotted using a matrix plot.

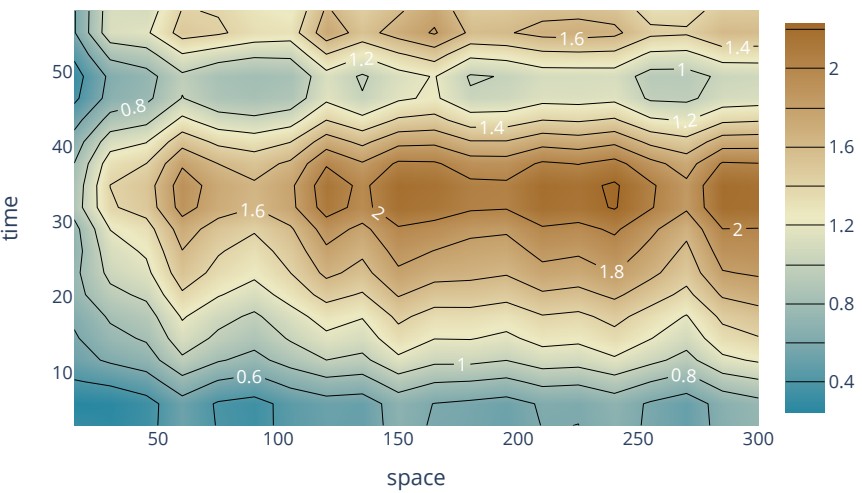

**Figure 7.** Contour plot of an experimental space-time variogram, without theoretical model. The shown variogram is from exactly the same instance as used for 6, without any modifications. The contours are calculated for the semi-variances (z-axis) and thus contain the same information as the scatter plot in figure 6.

To build a separable space-time variogram model, the two dimensions are first calculated separately. Non-separable space-time variogram models are not covered in SciKit-GStat. The two experimental variograms are called marginal variograms and relate to the temporal or the spatial dimension exclusively, by setting the other dimension's lag to zero. Finally, these two variograms are combined into a space-time variogram model. SciKit-GStat implements three models: the sum model, product model and product-sum model. For each of the marginal experimental variograms, a theoretical model is fitted, as described in

section 3.1. These two models $V_x(h)$ (spatial) and $V_t(t)$ (temporal) are then used to combine their output into the final model's return value $\gamma$. The space-time model defines how this combination is archived.

For the sum model, $\gamma$ is simply $V_x(h) + V_t(t)$. The product and product-sum models are implemented following De Cesare et al. (2002, equation (4), (6)).

## 4    Software implementation

This section focuses on the implementation of SciKit-GStat. It aims to foster an understanding of the most fundamental design decisions made during development. Thus, the reader will gain a basic understanding how the package works, where to get started and how SciKit-GStat can be extended or adjusted.



## 4.1 Main classes

SciKit-GStat is following an object-oriented programming (OOP) paradigm. It exports a number of classes, which can be
instantiated by the user. Common geostatistical notions are reflected by class properties and methods to relate the lifetime of
each object instance to typical geostatistical analysis workflows. At the core of SciKit-GStat stands the `Variogram` class for
variography. Other important classes are:

- `DirectionalVariogram` for direction dependent variography.

- `SpaceTimeVariogram` for space-time variography.

– `OrdinaryKriging` for ordinary kriging interpolations.

### 4.1.1 Variogram

The `Variogram` is the main class of SciKit-GStat and the only construct the user will interact with, in most cases. Each
instance of this class represents the full common analysis cycle in variography. That means, each instance will be associated
to a specific data sample and holds a fitted model. Other than other libraries, there is no abstraction of variogram models and
fitted models are not an entity of their own. If alternate input data (not parameters) is used a new object must be created.
This makes the transfer of variogram parameter onto other data samples a conscious action performed by the user and not a
side-effect of the implementation. At the same time parameters are mutable and can be changed at any time, which will cause
re-calculation of dependent results. While this design decisions makes the usage of SciKit-GStat straightforward, it can also
decrease performance. I.e., in SciKit-GStat, a variogram model is always fitted, even if only the experimental variogram is
used. This can be a downside, especially on large datasets. For cases where the full variogram instance is not desired or needed,
possible pathways are described in section 4.1.3 and 4.1.4, but the usage of `gstools` might be preferable in these cases.

    The second design decision for `Variogram` was interactivity. To take full advantage of OOP, every result, parameter and
plot is accessible as an instance attribute, property or method. This always clearly sets ownership and provenance relations for
data samples and derived results and properties, as there are no floating results that have to be captured in arbitrarily named
variables. Moreover, parameters that might be changed during a variogram analysis are implemented in a mutable way. A
substantial effort was made to store as few immutable parameters as possible in the instance. Thus, whenever a parameter is
changed at run-time, depending derived attributes and results will be updated. This convenient behavior for analysis comes at
the cost of performance. This is another major difference to the `gstools` library, in which the author assumes performance
to be a driving design decision.
To illustrate this as an example: When a variogram instance is constructed without further specifying the spatial model that
should be used, it will default to the spherical model. The instance is fitted to this model after construction and can be inspected
by the user i.e. by calling a plot method. The user wants to check out another semi-variance estimator, such as the Cressie-
Hawkins estimator, because there are a lot of outliers in the dataset. Changing the estimator is as easy as setting the literal

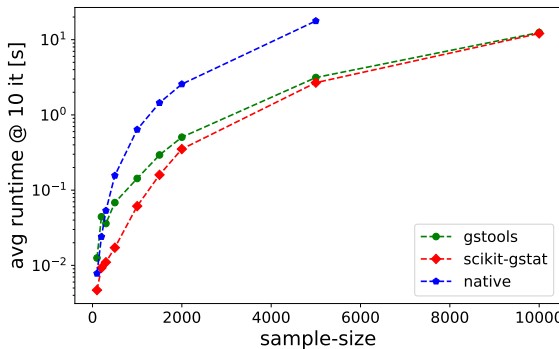

**Figure 8.** Benchmark test for estimating an experimental variogram. For each sample size, the mean runtime of ten repetitions is shown. The experimental variogram was calcualted with a native Python implementation (blue), `gstools` (green) and SciKit-GStat (red).

estimator name to the estimator property of the variogram. The experimental variogram will instantly be dropped and re-
calculated as well as all depending parameters, such as the variogram parameters. The spherical model is fitted a second time
now. The user might then realize that a spherical model is not suitable and can simply change the model attribute, i.e. to the
Matérn model. As a direct effect, the variogram parameters are dropped again, as they are once again invalidated, and a new
fitting procedure is invoked. This behavior is extremely convenient, as it is easy, interactive, expressive and instant. But it is also
slow, as i.e. the theoretical model had been fitted three times, before the user even looked into it. To add some context to *slow*
calculations, an experimental variogram estimation run-time test[3] has been performed (figure 8). One can see, that SciKit-GStat
and gstools are very comparable in this case and both significantly faster than a native Python implementation, especially for
larger datasets. Note the log-scaled y-axis, indicating differences of magnitudes for larger sample sizes. Interactively adjusting
variogram parameters will invoke additional calculations of given run-times.

Although most attributes are mutable, they use common data types in their formulation. This enables the user to intercept the
calculation at any point using either primitive language types or `numpy` data types, which are most accepted by the scientific
community as the prime array and matrix data types. Thus, there is no need for the user to learn about custom data-, parameter
or result structures using SciKit-GStat.

### 4.1.2 distance lag classes

Possibly the most crucial step to estimate a suitable variogram is the binning of separating distances into distance lag classes.
In some parts, SciKit-GStat also includes information theoretic methods. Here, to calculate the basic measure, Shannon En-
tropy (Shannon, 1948), the input data has to be binned to calculate empirical non-exceeding probabilities. To distinguish the

---

[3]This only tests the estimation of the experimental variogram and does not test any other functionality. I.e. kriging implementations in gstools are substan-
tially faster than in SciKit-GStat. The test was not performed in an isolated environment, but repeated several times.





information theoretic binning from the procedure of binning separating distances into classes, I will refer to the latter as *lag classes*. In the literature, lag classes are commonly referred to as bins, lags, distance lags or distance bins.

SciKit-GStat implements a large number of methods to form lag classes. They can be split into two groups: some are
adjusting class edges to fit the requested amount of lag classes. The other group will adjust the number of lag classes to fit other, statistical properties of the resulting lag classes. All methods can be limited by a maximum lag. This is a hyper-parameter, that can be specified by the user, but is not set by default. There are various options for the maximum lag. The user can set the parameter by an absolute value, in coordinate units and larger than one. Alternatively, a number between 0 and 1 can be set. Then, the `Variogram` class will set the maximum lag to this share of the maximum pairwise distance found in the distance
matrix. I.e. if 0.5 is used, the maximum lag is set to half of the largest point pair distance found. Note, that this is not a median value. Finally, a string can be set as maximum lag. This can either request the arithmetic mean or the median value of the distance matrix as the maximum lag. Typical values from geostatistical textbooks are the median or 60% of the maximum lag (value of 0.6 in SciKit-GStat).

The default behavior is to form a given amount of equidistant lag classes, from 0, to the maximum lag distance. This
procedure is used in the literature in almost all cases (with different max-lags), and is thus, a reasonable default method. Another procedure, takes the number of lag classes and will form lag classes of uniform size. That means, each lag class will contain the same amount of point pairs and, thus be of varying width. This procedure can be explicitly useful to avoid empty lag classes, which can easily happen for equidistant lag classes. Another advantage is that the calculation of semi-variance values will always be based on the same sample size, which makes the values statistically more comparable. These advantages
come at the cost of less comparable lag classes. Care must be applied when interpreting lag-related variogram properties such as the effective range. There might be lag ranges that are supported by only a very small amount of actual lag classes.

The next group of procedures use common methods from histogram estimation to calculate a suitable amount of lag classes. This is carried out, either directly, or by estimating the lag class width and deriving the amount of classes needed from this. The first option is to apply Sturge's rule (Scott, 2009) as shown in equation (7):

$$n = log_2(s+1) \tag{7}$$

Where $s$ is the sample size and $n$ is the number of lag classes. This rule works good for small, normally distributed distance matrices, but often yields too small $n$ for large datasets.

Similar to Sturge's rule, the square-root rule estimates the number of lag classes as given in equation (8):

$$n = \sqrt{(s)} \tag{8}$$





This rule is not recommended in most cases. It comes with similar limitation as Sturge's rule but in contrast, it usually yields too large $n$ for large $s$. The main advantage of this rule is that it is computationally by far the fastest of all implemented rules. Scott's rule (Scott, 2010) does not calculate $n$ directly, but rather $h$, the optimal width for the lag classes using equation (9):

$$h = \sigma \left( \frac{24 * \sqrt{\pi}}{s} \right)^{\frac{1}{3}} \tag{9}$$

Where $\sigma$ is the standard deviation of $s$. By taking $\sigma$ into account, Scott's rule works good for large datasets. It's application

does not work very well on distance matrices with outliers, as the standard deviation is sensitive to outliers.

If Scott's rule does, due to outliers, not yield suitable lag classes, the Freedman-Diaconis estimator (Freedman and Diaconis, 1981) can be used. This estimator is similar to Scott's rule, but makes use of the inter-quartile range as shown in equation (10):

$$h = 2 \frac{IQR}{s^{1/3}} \tag{10}$$

The inter-quartile range ($IQR$) is robust to outliers, but in turn the Freedman-Diaconis estimator usually estimates way too

many lag classes for smaller datasets. The author cannot recommend to use it for distance matrices with less than 1000 entries. Finally, Doane's rule (Doane, 1976) is available. This is an extension to Sturge's rule, that takes the skewness of the sample into account. This makes it especially suitable for smaller, non-normal datasets, where the other estimator do not work very good. It is defined as given in equation (11):

$$n = 1 + \log_2(s) + \log_2 \left( 1 + \frac{|g|}{k} \right)$$
$$g = E \left[ \left( \frac{x - \mu_g}{\sigma} \right)^3 \right]$$
$$k = \sqrt{\frac{6(s - 2)}{(s + 1)(s + 3)}} \tag{11}$$

Here, $g$ is the skewness, $\sigma$ is the standard deviation, $\mu_g$ is the arithmetic mean and $x$ is each element in $s$.

All rules that calculate the number of lag classes use the `numpy` implementation of the respective methods (van der Walt et al., 2011).

All histogram estimation methods given above just calculate the number of lag classes. The resulting classes are all equidistant, except for the first lag class, which has 0 as a lower bound, instead of $min(s)$.

Finally, SciKit-GStat implements two other methods. Both are based on a clustering approach and need the number of lag classes to be set by the user. The distance matrix is clustered by the chosen algorithm. Depending on the clustering algorithm, the cluster centers (centroids) are either estimates of high density or points in the value space, where most neighboring values have the smallest mean distance. Thus, the centroids, are taken as a best estimate for lag class centers. Each lag class is then





formed by taking half the distance to each sorted neighboring centroid as bounds. This will most likely result in non-equidistant
lag classes.

The first option is to use the K-Means clustering algorithm, which is maybe the most popular clustering algorithm. The method
is often attributed to MacQueen et al. (1967), but there are thousands of variations and applications published. The implemen-
tation of K-Means used in SciKit-GStat is taken from `scikit-learn` (Pedregosa et al., 2011). One important note about
K-Means clustering is, that it is not a deterministic method, as the starting points for clustering are taken randomly. In practice
this means, that exactly the same `Variogram` instantiated twice can result in different lag classes. Experimental variograms
are very sensitive to the lag classes. In some unsystematic tests undertaken by the author, the variations in lag class edges
could be as large as 5% of the distance matrix range, which would result in substantially different experimental variograms.
Thus, the decision was made to seed the random start values. For this reason, the K-Means implementation in SciKit-GStat is
deterministic and will always return the same lag classes for the same distance matrix. The downside is, that the clustering loses
some of it's flexibility and can't be cross-validated. Furthermore, the K-Means will find one set of lag classes, not necessarily
the best one. However, the user can still calculate lag class edges externally, using K-Means, and pass the edges explicitly to
the `Variogram` class.

The other clustering algorithm is a hierarchical clustering algorithm (Johnson, 1967). These algorithms group values together
based on their similarity. SciKit-GStat uses an agglomerative clustering algorithm, which uses Ward's criterion (Ward Jr and
Hook, 1963) to express similarity. Agglomerative algorithms work iteratively and deterministic, as at first iteration each value
forms a cluster on its own. Each cluster is then merged with the most similar other cluster, one at a time, until all clusters are
merged, or the clustering is interrupted. Here, the clustering is interrupted as soon as the specified number of classes is reached.
The lags are then formed similar to the K-Means method, either by taking the cluster mean or median as center. Ward's crite-
rion defines the one other cluster as the closest, that results in the smallest intra-cluster variance for the merged clusters. That,
finally results in slightly different lag class edges than K-Means. The main downside of the agglomerative clustering is that it
is by far the slowest method. In some cases, especially for larger datasets, the clustering took longer than the full workflow to
estimate a variogram and fit a theoretical model, by magnitudes.

The implementation follows `scikit-lean` (Pedregosa et al., 2011). Using the `AgglomerativeClustering` class with
the `linkage` parameter set to `'ward'`.

One method of utilizing clustered lag classes is to compare the K-Means lag edges with the default settings. The idea is
to minimize the deviation of both while searching a suitable amount of classes. This combines the advantages of K-Means,
while yielding equidistant lag classes, that have the best match to clustered centroids. SciKit-GStat makes that possible, while
leaving the interpretation to the user.

Another option available is called *stable entropy*. This is a custom optimization algorithm, that has not been reported before.
The algorithm takes the number of lag classes as a parameter and starts with the equidistant lag classes as a initial guess
for optimization. It seeks to adjust bin edges until all lag classes show a comparable Shannon entropy. The Shannon entropy
is calculated using equation (15), with a static binning created analogous to equation (8), the square-root rule for histogram





estimation. The lag classes are optimized by minimizing the absolute deviation in Shannon entropy, at a maximum of 5000

iterations. The algorithm uses the Nelder-Mean optimization (Gao and Han, 2012) implemented in `scipy` (Jones et al., 2001–

). As the Shannon entropy is a measure of uncertainty based on information content, it is expected to yield statistically robust

lag classes. At the same time it is expected to show the same limitations as the uniformly sized lag classes, such as a potentially

difficult interpretation of variogram parameters.

### 4.1.3 sub-module: estimators

SciKit-GStat implements a number of semi-variance estimators. It includes all semi-variance estimators that are commonly

used in the literature.

**numba support**: The `numba` package offers function decorators, that enable just-in-time compilation of Python code. Although there are ways to compile code even more effectively (i.e. `cython, nuitka` packages), `numba` comes at zero implementation overhead and fair calculation speed ups. The `numba` decorator is implemented for the matheron, cressie, entropy and genton estimators. For the other estimators, the just-in-time compilation adds more compiling overhead, than a compiled

version actually gains performance on reasonable data sample sizes. The main reason is, that the remaining estimators are already covered mathematically by a `numpy` function, which are in most cases already implemented in a compiled language.

**matheron**: The `matheron` function implements the Mathéron semi-variance $\gamma$ (Matheron, 1963). This estimator is so commonly used, that it is often referred to just as *semi-variance* and thus the obvious default estimator in SciKit-GStat. It is defined in equation (1).

**cressie** implements the Cressie-Hawkins estimator $\gamma_c$ (Cressie and Hawkins, 1980). As given in equation (12):

$$2\gamma_c(h) = \frac{\left(\frac{1}{N(h)} \sum_{i=1}^{N(h)} |Z(s_i) - Z(s_{i+h})|^{0.5}\right)^4}{0.457 + \frac{0.494}{N(h)} + \frac{0.045}{N^2(h)}} \tag{12}$$

Where $N(h)$ is the number of point pairs $s, s_i$ at separating lag $h$ and $Z(s)$ is the observation value at $s$.

**dowd** implements the Dowd estimator $\gamma_D$ (Dowd, 1984). As given by equation (13):

$$2\gamma_D(h) = 2.198 * median(Z(s_i) - Z(s_{i+h}))^2 \tag{13}$$

This estimator is based on the median value of all pair-wise differences $s_i, s_{i+h}$ separated by lag $h$, where $Z(s)$ is the observation value at location $s$. Thus, the Dowd estimator is very robust to outliers in the pair-wise differences and very fast to calculate.

**genton** implements the Genton estimator $\gamma_G$ (Genton, 1998). As given by equation (14):

$$\gamma_G(h) = 2.2191\{|Z_i(s_i) - Z_j(s_j)|; i < j\}_{\left(\frac{k}{q}\right)}$$
$$k = \binom{[N(h)/2] + 1}{2}$$
$$q = \binom{N(h)}{2} \tag{14}$$



Where the pair-wise differences $Z(s_i), Z(s_j)$ at separating lag $h$ are only used if $i < j$. The n[th] percentile is calculated from $k, q$, which are both binomial that only depend on the number of point pairs $N(h)$. The implementation in SciKit-GStat simplifies the application of the equation by setting $\frac{k}{q} := 0.25$ for $N(h) >= 500$. This avoids the necessity to solve very large binomials at negligible errors, as $\lim_{N(h) \to \infty} \frac{k}{q} = \frac{1}{4}$. The author has found the Genton estimator to yield a reasonable basis for variogram estimation in many environmental applications (a personal, maybe biased observation). However, calculating the

binomials requires some time. Especially if there are a lot of lag classes and a considerable amount of them does not fulfill the $N(h) >= 500$ constraint, it will slow the calculation down by many magnitudes compared to the other estimators.

**minmax** implements a custom estimator. The author is not aware of any publication of this estimator. It was introduced during development, as it has quite predictable statistical properties. However, I am also not aware of any useful practical applications of this estimator and can thus not recommend using it in typical geostatistical analysis workflows.

The MinMax estimator divides the value range of pairwise differences by their mean value.

**entropy**: Is an implementation of the Shannon Entropy $H$ (Shannon, 1948) as a semi-variance estimator. An successful application of Shannon Entropy as a measure for similarity in dependence of spatial proximity has been reported by Thiesen et al. (2020). The Shannon Entropy is defined with equation (15):

$$H(h) = -\sum_{i=1}^{N(h)} p_i log_2(p_i)$$

(15)

Where $p_i$ is the empirical exceeding probability of $Z(s_i) - Z(s_{i+h})$ for each separating lag $h$. To calculate the empirical probabilities of occurrence, a histogram of all pairwise differences is calculated. This histogram has evenly spaced bin edges and the user can set the amount of bins as a hyper-parameter to entropy. Alternatively, the bin edges can be set explicitly. One has to be aware that the Shannon Entropy relies on a suitable binning of the underlying data. This might need some preliminary examination of $Z(s_i) - Z(s_{i+1})$, which is readily accessible as a property. It is highly recommended to use exactly

the same bin edges for all separating distances $h$ needed to process a single variogram. Otherwise the entropy values and their gradient over distance is not comparable and the whole variogram analysis turns meaningless.

Finally, it is possible to use custom user-defined functions for estimating the semi-variance. The function has to accept a one dimensional array of pair-wise differences, as these are already calculated by the Variogram class. The return value must be a single floating point value. This can either be the primitive Python type or a 64bit numpy float. The given function is

finally mapped to all separating distance lags automatically, thus there is no need to implement any overhead, such as sorting or grouping, by the user. This empowers users with little or no experience in Python to define new semi-variance estimators as only the mathematical description of the semi-variance is needed as Python code.

### 4.1.4 sub-module: models

SciKit-GStat implements a number of theoretical variogram models. The most commonly used models from literature are
available. However, during researching theoretical models, the author brought an almost limitless number of models, or vari-





ations thereof to light. Thus, the process of implementing new models was eased as far as possible, instead of implementing anything that could be useful. Any variogram model function (implemented and custom) will receive the *effective range* as a function argument and is fitted using it. In case the mathematical model of a variogram function uses the range parameter, one has to implement the conversion into the model function as well.

The core design decision for SciKit-GStat's theoretical variogram models was to implement a decorator, that wraps any model function. This decorator takes care of handling input data and aligning output data. Thus, the process of implementing new variogram models is simplified to writing a function that maps a single given distance lag to the corresponding semi-variance value.

Each model will receive the three variogram parameter effective range, sill and nugget as function arguments. The nugget

is implemented as a optional argument with a default value of zero, in case the user disables the usage of a nugget in the `Variogram` class. Custom variogram models have to reflect that behavior.

**spherical** is the implementation of the spherical model, which is one of the most commonly used variogram models. Thus, the spherical variogram model is the default model, in case the user did not specify a model explicitly. The model equation is taken from Burgess and Webster (1980) and given in equation (16):

$$\gamma(h) = \begin{cases} b + C_0 * \left(1.5 * \frac{h}{a} - 0.5 * \frac{h}{a}^3\right) & h < a \\ b + C_0 & h \geq a \end{cases}$$

$$a := r \tag{16}$$

Where $h$ is the distance lag and $b, C_0, a$ are the variogram model parameters: nugget, sill and range. The range of a spherical model is defined to be exactly the effective range $r$.

**exponential** is the implementation of the exponential variogram model. The implementation is taken from Journel and Huijbregts (1976) and given in equation (17):

$$\gamma(h) = b + C_0 * \left(1 - e^{-\frac{h}{a}}\right)$$

$$a = \frac{r}{3} \tag{17}$$

Where $h$ is the distance lag and $b, C_0, a$ are the variogram model parameters: nugget, sill and range. For the exponential model, the effective range $r$ is different from the variogram range parameter $a$.

**gaussian** is the implementation of the Gaussian variogram model. The implementation is taken from Journel and Huijbregts (1976) and given in equation (18):

$$\gamma(h) = b + c_0 * \left(1 - e^{-\frac{h^2}{a^2}}\right)$$

$$a = \frac{r}{2} \tag{18}$$





Where $h$ is the distance lag and $b, C_0, a$ are the variogram model parameters: nugget, sill and range. For the Gaussian model, the effective range $r$ is different from the variogram range parameter $a$. In SciKit-GStat, the conversion from effective range to range parameter is implemented as shown in equation (18). However, the author is aware of other implementations in literature. The package does not allow to somehow switch the conversion and the user has to implement a new Gaussian model, in case
another conversion is desired.

**cubic** is the implementation of the cubic variogram model. The implementation is taken from Montero et al. (2015) and given in equation (19):

$$\gamma(h) = \begin{cases} b + C_0 * \left[ 7 * \left( \frac{h^2}{a^2} \right) - \frac{35}{4} * \left( \frac{h^3}{a^3} \right) + \frac{7}{2} * \left( \frac{h^5}{a^5} \right) - \frac{3}{4} * \left( \frac{h^7}{a^7} \right) \right] & h < a \\ b + C_0 & h \geq a \end{cases} \qquad a := r \tag{19}$$

Where $h$ is the distance lag and $b, C_0, a$ are the variogram model parameters: nugget, sill and range. For the cubic model, the
effective range $r$ is exactly the variogram range parameter $a$.

**matern** in the implementation of the Matèrn variogram model. The implementation is taken from Zimmermann et al. (2008) and given in equation (20):

$$\gamma(h) = b + C_0 \left( 1 - \frac{1}{2^{v-1}\Gamma(v)} \left( \frac{h}{a} \right)^v K_v \left( \frac{h}{a} \right) \right)$$
$$a = \frac{r}{2} \tag{20}$$

Where $h$ is the distance lag, $\Gamma$ is the gamma function and $b, C_0, a$ are the variogram model parameters: nugget, sill and range.
Additionally, the Matérn model defines a fourth model parameter $v$, which is a smoothness parameter. For the Matérn model, the effective range $a$ is a fraction of the variogram parameter range $r$.

**stable** is the implementation of the stable variogram model. The implementation is taken from Montero et al. (2015) and given in equation (21):

$$\gamma(h) = b + C_0 * \left( 1. - e^{-\frac{h}{a}^s} \right)$$
$$a = \frac{r}{3^{s^{-1}}} \tag{21}$$

Where $h$ is the distance lag and $b, C_0, a$ are the variogram model parameters: nugget, sill and range. Additionally, the stable model has a shape parameter $s$. The effective range of the variogram is a fraction of the variogram range parameter, dependent on this shape. Generally, the effective range will increase with larger shape values.

**harmonize** is an implementation, that is rather uncommon in geostatistics. It is based on the idea of monotonizing a data sample into a non-decreasing function. That means, there is no model fitting involved and the procedure bypasses all related
steps. A successful application in geoscience was reported by Hinterding (2003). For SciKit-GStat, the more generalized approach of isotonic regression (Chakravarti, 1989) was used which is already implemented in `scikit-learn` (Pedregosa et al., 2011).





Note, that a harmonized model might not show an effective range, in which cases the library will take the maximum value as the effective range for technical reasons. Thus, the user has to carefully double-check harmonized models for their geostatistical

soundness. Secondly, the harmonized model cannot be exported to `gstools`, which makes it unavailable for most Kriging algorithms.

### 4.1.5 Fitting theoretical models

As soon as an estimated variogram is used in further geostatistical methods, such as kriging or field simulations, it is necessary to describe the experimental, empirical data by a model function of defined mathematical properties. I.e., for kriging,

a variogram has to be monotonically increasing and positive definite. This is assured, by fitting a theoretical model to the experimental data. The models available in SciKit-GStat are described in section 4.1.4.

Fitting the theoretical model to the experimental data is crucial, as any uncertainty caused by this procedure will be propagated to any further usage of the variogram. Almost any geostatistical analysis workflow is based on some kind of variogram and hence, the goodness of fit will influence almost any analysis. The `Variogram` class can return different parameters to

judge the goodness of fit, among other the coefficient of determination, root-mean squared error and mean squared error. Beyond a direct comparison of experimental variogram and theoretical model, the `Variogram` class can run a leave-one-out cross validation of the input locations to assess the fit based on kriging. As the experimental values and their modeled counterparts are accessible for the user at all times, implementations of any other desired coefficient are straightforward.

When fitting the model, SciKit-GStat implements four main algorithms, each one in different variations. A main challenge

of fitting a variogram model functions is, that closer lag classes result in higher kriging weights and are therefore of higher importance. A variogram model that might show a fair overall goodness of fit, but is far off on the first few lag classes, will result in poorer kriging results, than an overall less well fitted model that hits the first few lags perfectly. On the other hand, emphasizing the closer lags is mainly done by adjusting the range parameter. The only other degree of freedom for fitting the model is then the sill parameter. Thus, if the modeling of the closer lag classes is put too much into focus, this happens at the

cost of missing the experimental sill, which is basically the sample variance, in case the nugget is set to zero. If the nugget is not zero, a insufficient sill will change the nugget to sill ratio and one might have reject the variogram at all. A kriging interpolation of reasonable range is able to reproduce the spatial structure of a random field, but if the sill is far off, the interpolation is not able to reproduce the value space accordingly and the estimations will be inaccurate. In the extreme case of a pure nugget variogram model, kriging will only estimate the sample mean (which is the correct behavior, but not really useful). Thus, the

fitting of a model has to be evaluated carefully by the user and SciKit-GStat is aiming to support the user with this.

A procedure that is frequently used to find optimal parameters for a given model to fit a data sample is *least squares*. These kinds of procedures find a set of parameters, that minimize the squared deviations of the model to observations. A robust, widely spread variant of least squares is the Levenberg-Marquardt algorithm (Moré, 1978). It is a robust and fast fitting algorithm that yields reasonable parameters in most cases. However, Levenberg-Marquardt is an unbounded least-squares algorithm, meaning

that value space for the parameters can neither be limited, nor constrained. In the specific case of variogram model fitting, there





are a number of assumptions that actually do constrain the parameter space. Thus, in some occasions, Levenberg-Marquardt is failing to find optimal parameters, as it is searching parameter regions, that would not be valid variogram parameters, anyway. The implementation for Levenberg-Marquardt least squares is taken from the `scipy` package (Jones et al., 2001–).

Another least-squares approach is Trust-Region Reflective (TRF) (Branch et al., 1999). A major difference to Levenberg-Marquardt is that TRF is a bounded least-squares algorithm. That means, the `Variogram` class can set lower and upper limits for each of the parameters. Thus, the TRF is, from what I can say, always finding suitable parameters and is therefore the default fitting method in SciKit-GStat.

The adjustable variogram model parameters are the effective range, sill, nugget, if used, and a shape parameter for the Matèrn and stable model. The lower bound for all parameters is zero, as all parameters have to be positive by definition. The upper

bounds can also be defined for all parameters. The effective range is bounded to the maximum lag, or largest separating distance observed, if the maximum lag was not specified by the user. The sill is bounded by the largest semi-variance value that was estimated for the experimental variogram. As nugget and sill effectively sum up to sample variance, it consequently has to be smaller than any individual semi-variance value. The same has to hold for the nugget, due to the implementations given in section 4.1.4. For technical reasons, the sill must not be 0. The nugget has the same upper bound as the sill, as TRF does not

take constrains, only parameter bounds (a constraint would put a dependency of one parameter to the other into the algorithm, which would be the more appropriate handling here).

The implementation for Trust-Region Reflective least squares is taken from the `scipy` package (Jones et al., 2001–).

The third fitting method is a maximum likelihood approach. The theoretical model is fitted to the experimental data by minimizing the negative log-likelihood of the variogram parameters. Each of the parameters samples from a normal distribution

with the last parameters predictions mean and standard deviation as first and second moment. In the current implementation, an unbounded and unconstrained Nelder-Mead solver (Gao and Han, 2012) is used to minimize the log-likelihood function. The implementation is taken from `scipy` (Jones et al., 2001–). For rare cases where this solver is not able to find valid variogram parameters, the SLSQP (Kraft et al., 1988) algorithm can be used. It is substantially slower but more flexible and will search the best parameters in a valid parameter space only. Without having performed a systematic testing beyond unit-tests for the

maximum likelihood option, it seems like the maximum likelihood estimation often struggles with larger nugget values and does not find optimal variogram parameters.

The last option is not an algorithm. The `Variogram` class has the ability to directly take the variogram parameters from the user as hyper-parameters. In these cases the class will bypass the fitting procedures and just set the user input as fitting coefficients. This is convenient for cases where the user receives the parameters externally. It is also possible to switch to

custom fitting, after another algorithm had already been used. This can be helpful to fine-tune automatically fitted parameters. On the other hand, the implementation does also bypass all checks and constrains made to the parameter space and the user could i.e. pass invalid values. An example is a negative nugget value, which is mathematically applicable (there is i.e. no runtime error), but does not make any sense from a geostatistical point of view. Ensuring variogram validity is completely in the responsibility of the user in these cases.

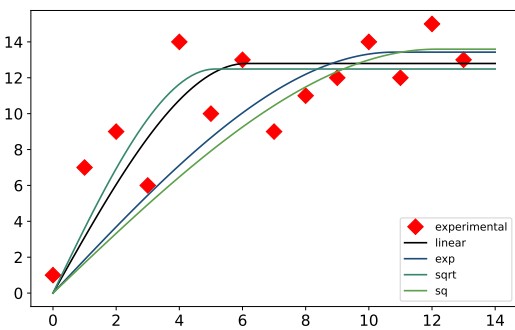

**Figure 9.** Red squares show a sample experimental variogram (values are made up) with four different spherical variogram models. All four models are fitted using Trust-Region Reflective fitting procedure and distance depended weights. The weights are **linear** decreasing with distance (blue line), decreasing by the **square-root** of the normalized distance (green line), the **squared** normalized distance (red line) and decreasing by **e-function**, as shown in equation (22) (yellow line).

All fitting mechanisms except for the manual fit, can be further refined by setting an array of fitting weights. This enables the user to focus only a few lag classes for fitting and achieve a higher goodness of fit on specific lags. The weights are, following the logic of `scipy`, actually not weights, but uncertainties. Thus, if one has only weights available, their inverse has to be used. It is possible to pass a numeric value array to the `Variogram` class, that has to be of same length as the number of lag classes. If not set, the `Variogram` will equally weight all lag classes. In most other cases the user will want to apply decreasing

weights with increasing separating distance, to put more focus on the first few lag classes. SciKit-GStat conveniently includes a number of functions, that calculate an uncertainty array that will effectively apply decreasing weights.

The first option is a linear decrease of weights with increasing lags. The second option uses the square-root of the the normalized lag as an approximation. The third option uses the inverse of the normalized lag squared as a weight. This results in completely neglecting any lag class but the first two or three, depending on the total amount. The last function applies an exponential

function as given by equation (22):

$$\frac{1}{w} = e^{lag_n^2} \tag{22}$$

Where $w$ is the calculated weight and $lag_n$ the normalized lag.

All four distance-dependent weighting functions are compared in figure 9. All four functions show very comparable coefficients of determination, calculated over all lag classes. That means, the four models describe the experimental variogram equally

well. It is now up to the user to decide which one to use. SciKit-GStat does not apply any of these distance weighting functions automatically. This example illustrates, how important it is to examine experimental variograms and the many possibilities how one can capture its properties in a theoretical model, before approaching more complex geostatistical methods like kriging or field generation. Otherwise, the choice of model and model parameters might seem arbitrary. To illustrate this, the four models resulting solely from a different weighting of the lag classes for fitting (figure 9) were used to generate a random field.





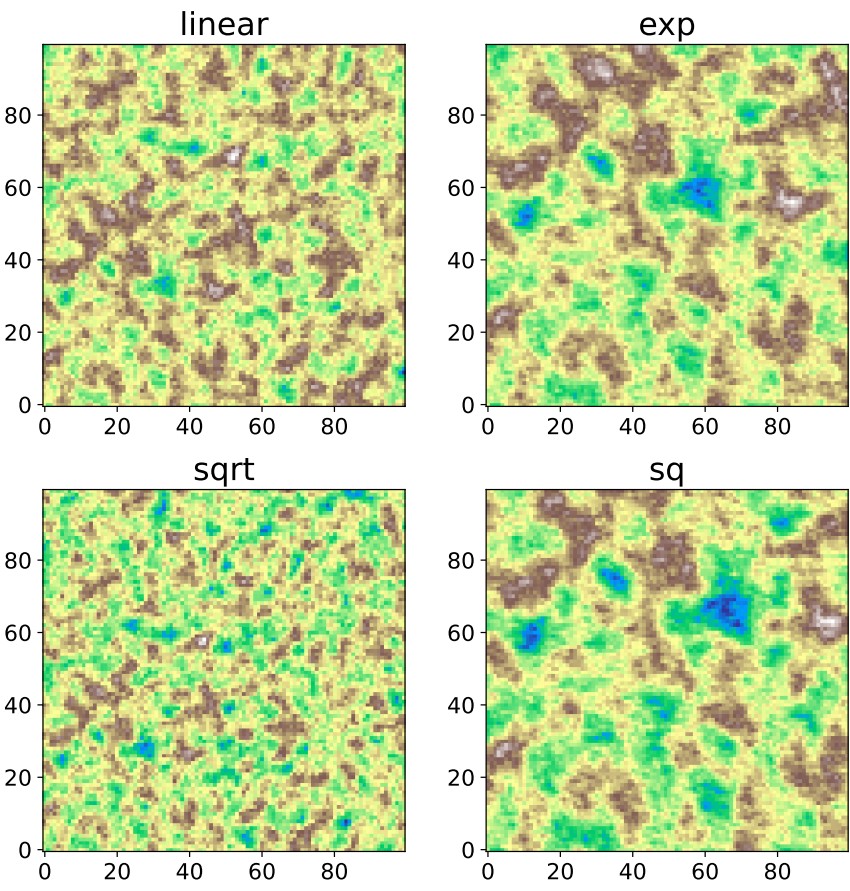

**Figure 10.** Four random fields generated using the same seed for randomization, which results in exactly the same field for same input. The only differing input parameter is the automatic distance weighting function that was used for fitting the theoretical variogram model.

The generation of the random field was seeded with a fixed value, in order to create reproducible results and hence the only difference in the fields originates from the choice of weighting function (figure 10). Finally, the fitting of variogram models is usually neither exposed to the user (as sometimes even not the variogram itself), nor does the user have control on the internals of fitting. In the shown example (figure 10), only one parameter that influences fitting was changed, and that shows dramatic effects. SciKit-GStat seeks to give the user more options to assume control over this important step. Each of the other options

for fitting might well produce similar dramatic changes in field generation. Hence, it is so important to assess automatically derived fitting results, because finally it should be up to human interpretation whether a variogram should be used or not.

Another predefined possibility to determine weights for fitting is information theory. Unlike the other functions, this option is not based on an inverse of weights. The information theory based weighting option calculates the uncertainties directly, by using the Shannon Entropy (Shannon, 1948). It is calculated for the empirical distribution of point pairs within each distance





lag class. This will link the weight during fitting directly to the information content of that lag. From a practical point of view, the resulting weights are usually closer to uniform weights, than the distance dependent weights. For the distance weighted procedures, the larger lags are almost completely ignored. With the information theoretic approach this will only happen for very thin populated lag classes.

### 4.1.6   Directional variograms

Directional variograms can be estimated in SciKit-GStat using the `DirectionalVariogram` class. It inherits from `Variogram`, making all its properties and methods available. Only methods that actually work on the distance matrix are re-implemented to intercept calculations with a spatial filter. This let's the user interact with the class as learned with the base class, focusing only on the differences between a directional variogram calculation and a classic.

  `DirectionalVariogram` only overwrites one internal method and one property of the base class. This is the logic
assigning the correct lag group to each point pair calculated and then deriving the lag bin edges from this. In both cases, point pairs are filtered by their orientation, before the calculation is continued. This way, `DirectionalVariogram` only adds necessary calculations steps and the base class does not have to handle data, information or logic (such as point pair orientation) that does not affect the classic calculation. This conscious design decision leaves the code as readable as possible to make contributions easier for others.

Three new attributes are introduced, that can be set by the user. For all three parameters SciKit-GStat retains the name, implementation and usage as close to Montero et al. (2015) as possible.

The **azimuth** of the directional variogram is the direction for which the directional variogram will be calculated. It is given in degrees as a counter-clockwise deviation from the coordinate x-axis (which will be East in most cases). The **tolerance** is an angle in degrees, which defines the limit at which a deviation from the azimuth is still acceptable. Only these point pairs will
be taken into account, which orientation as calculated with equation (6) are within the tolerance of the azimuth. The tolerance defaults to 45 degrees.

As the tolerance is given in degrees, the absolute deviations in the unit of the coordinate system can be quite considerable for larger separating distances. Therefore, it is possible to set a **bandwidth**. This parameter limits the maximum acceptable perpendicular distance from the azimuth vector in coordinate units and default to the 33% percentile of the distance matrix. It
can be set as a percentile or as a absolute limit, in coordinate units.

  Apart from the basic hyper-parameters that define a directional variogram, there are different implementations, how to apply them. SciKit-GStat denotes these implementations as *directional models* and implements two different.

The default *triangle* model is applying the three directional parameters as most often reported in literature (Montero et al., 2015), by constructing a triangle in the direction of the azimuth using the tolerance as a opening window. For larger distances,
the triangle is bounded by the bandwith and turned geometrically into a rectangle.

The unbounded version of the *triangle* model is called *compass*, which simply ignores the bandwidth parameter. Thus, it will only restrict point pairs to be oriented into a specific direction.

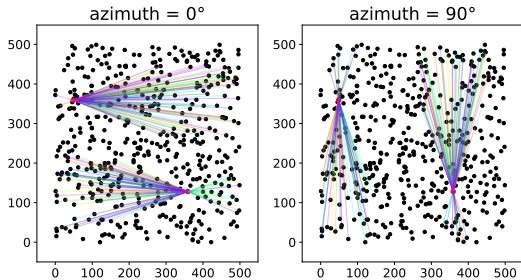

**Figure 11.** Pair Field plot of two directional variograms. The plot was created with exactly the same two directional variogram instances as used in figure 5. Both figures show the network graph for two observation points (index 42 and 170 in the sample file), in both directions of the variogram. The lines connect all point pairs that were taken into account for these two points. The line colors have no meaning and are just included for visual reasons.

For convenience and to further inspect the point pairs which are actually taken into account, there is an additional auxiliary plotting method. This plots a network graph for all input locations with a edge for each point pair that will be taken into account

765 for calculation (figure 11). Unlike other network graphs, the vertices keep their real locations in the coordinate space to identify specific input data points. A plot like this can be helpful to specify reasonable azimuth and tolerance values, which will highly impact the result.

### 4.1.7 Spatio-temporal Variogram

For calculating spatio-temporal variograms, SciKit-GStat has a class called `SpaceTimeVariogram`. Other than the `DirectionalVariogram`

770 class, `SpaceTimeVariogram` does not inherit from `Variogram`, but is an independent class. For a spatio-temporal variogram, any processing step is not only dependent on a spatial lag, but also on a temporal lag. This actually changes the function signatures for almost all methods and, thus it was decided to re-implement the whole class without any inheritance. Nevertheless, `SpaceTimeVariogram` and `Variogram` share attribute and method names wherever possible.

At the core of all implemented theoretical variogram methods for the spatio-temporal variogram is the estimation of two

775 marginal variograms. The class will estimate a *temporal* and a *spatial* marginal variogram. These are both instances of the `Variogram` class. The spatio-temporal models themselves expect both marginal variograms as an attribute.

Finally, the `SpaceTimeVariogram` implements a rich plotting method. It can plot the the experimental spatio-temporal variogram and the fitted theoretical model as a 3D or 2D plot (figure 6, 7). For 2D plotting, different plot types are implemented, i.e. a contour plot for semi-variance values. Both 2D and 3D plots are available. 3D plots allow for the user to interactively

780 rotate, pan and zoom the plot, enabling the user to inspect a spatio-temporal variogram. 2D plots are helpful for printed material.





### 4.1.8 sub-module: stmodels

SciKit-GStat implements three different theoretical spatio-temporal variogram models: the sum, product and sum-product model. In line with the `models` sub-module, the `stmodels` sub-module has a decorator functions to wrap the models. This decorator takes care of the data flow and leaves the implementation of the mathematical formula to the user, if custom models should be used.

In the following equations the marginal variograms represented by $\gamma_x, \gamma_t$ refer only to the spatial lag $h$ or temporal lag $t$, respectively. They are estimated and modeled as described in section 3.1 using any of the semi-variance estimators from section 4.1.3 and any model described in section 4.1.4.

**sum** is the implementation of the sum model. This is the most basic spatio-temporal model, a sum of a spatial marginal variogram $V_x(h)$ and a temporal marginal variogram $V_t(t)$ as shown in equation (23):

$$\gamma(h,t) = \gamma_x(h) + \gamma_t(t) \tag{23}$$

Where $\gamma_x, \gamma_t$ are the semi-variance estimations by the two marginal variograms and are not restricted to a specific semi-variance estimator or theoretical model.

The sum model provides an understanding of the idea and workflow of spatio-temporal models. However, it should not be used for real data in almost all cases. It assumes the covariance field to be isotropic across temporal and spatial dimensions. A situation which can be considered rarely true. Moreover, it might not be positive definitive, as required for variogram models (Myers and Journel, 1990; Dimitrakopoulos and Luo, 1994).

**product** is the implementation of the product model. The implementation is taken from De Cesare et al. (2002, equation (4), p.207) as shown in equation (24):

$$\gamma(h,t) = C_x * \gamma_t(t) + C_t * \gamma_x(h) - \gamma_x(h) * \gamma_t(t) \tag{24}$$

Where $C_x$ is the sill parameter of the spatial marginal variogram $\gamma_x(h)$ and $C_t$ is the sill of the temporal marginal variogram $\gamma_t(t)$.

**product_sum** is the implementation of the product-sum model. The implementation is taken from De Cesare et al. (2002, equation (6)) as shown in equation (25):

$$\gamma(h,t) = [k_1 C_T + k_2] * \gamma_x(h) + [k_1 C_s + k_3]\gamma_t(t) - k_1\gamma_x(h)x\gamma_t(t) \tag{25}$$

Here, $k_1, k_2, k_3$ are additional fitting parameters needed for the product-sum model. All three parameters need to be positive and may not be larger than any of the marginal sill parameters $C_x, C_t$.

### 4.1.9 Ordinary Kriging

SciKit-GStat implements a ordinary kriging algorithm. It is implemented following Montero et al. (2015) and can be used using the class `OrdinaryKriging`. The user needs to pass a instance of `Variogram` as a parameter. In a majority of other





Kriging implementations, the procedure accepts the observations and estimates a variogram automatically. Sometimes even as an internal processing step. For SciKit-GStat, the decision was made to focus on variogram estimation. The Kriging class should be seen as an auxiliary class to implement the full typical geostatistical analysis workflow. The user is encouraged to take a closer look on the variogram, utilizing all the plotting routines and descriptions, before passing it on to the Kriging class.

This should have a positive effect on geostatistical applications.

It must be noted that the `OrdinaryKriging` class is mainly implemented for cross validating Variogram models. It does not claim to be a very performing implementation of the Kriging algorithm. Nor is it implemented with the flexibility and analysis tools, the `Variogram` has. The author is also aware, that further kriging algorithms exist and ordinary kriging might not be the most useful one. Thus, SciKit-GStat is more focused on implementing interfaces to other libraries, that including

other kriging methods. Namely these are `gstools` and `pykrige`. To date, the two aforementioned libraries are aligned to each other, future `pykrige` iterations will implement `gstools` co-variograms. This will leave SciKit-GStat only with the need for a powerful interface to `gstools` to provide the full power of `pykrige` to SciKit-GStat users. The SciKit-GStat `Variogram` class has an interface function, that can instantiate any `gstools` Kriging algorithm from a SciKit-GStat variogram. More details on SciKit-GStat and `gstools` and their future co-existance are given in section 4.2.

## 4.2 SciKit-GStat and `gstools`

SciKit-GStat has three interfaces to `gstools`, all three implemented as instance methods of the `Variogram` class. The first option is to export the empirical variogram. This is the combination of the lag classes edges with the experimental variogram. The lag classes edges can optionally be shifted to the class centers, as this is the notation that `gstools` uses for empirical variograms. This interface is useful in case one of the many binning functions or semi-variance estimators was used, that is not

available in `gstools`.

The second, major, option is to translate the theoretical model into a fitted covariance model instance of `gstools`, which is their respective base class. With that in place, one can use the covariance model in conjunction with all the great methods available in `gstools`.

For the specific case of kriging, a third interface exports the variogram directly into a `gstools` kriging class instance. At

the time of writing, currently available kriging algorithms are simple kriging, ordinary kriging, universal or regression kriging, kriging with external drift and kriging the mean (Müller and Schüler, 2021).

Both libraries chose different avenues, how the user may interact with the library. For `gstools`, the user defines a covariance model and passes it to one of the rich set of geostatistical functions, which can be found in `gstools`. The user then captures the return value of the function and uses it for further development and analysis. In SciKit-GStat, as described in this

paper, the user rather instantiates one object and mutates it during the analysis.





## 5 Support, Application and Contribution

### 5.1 User support

Users are supported by a comprehensive documentation that includes API reference, installation instructions, getting started guide, a detailed user guide and tutorials. The user guide is written at the example of a lecture script. No geostatistical prior knowledge is necessary. Only some limited experience in Python and basic knowledge of univariate statistics is advantageous. Additionally, the user guide includes a number of technical notes, that discuss some specialities of SciKit-GStat in great detail.

SciKit-GStat is managed and hosted on Github under a MIT license. For technical problems, questions and feature requests, the Github issues ticketing system is used. To date any issues arising have been processed by the author himself. As some of the raised issues discussed fundamental geostatistical principles and basic applications of SciKit-GStat, these closed issues are also a valuable resource for new users to SciKit-GStat as well as geostatistics. The evaluation of these issues was taken into account for compiling the user guide.

To use SciKit-GStat in production environments and also for rapid installation, a docker image is offered. The Dockerfile is also included into the SciKit-GStat repository ,and therefore, also distributed under MIT license enabling users to adapt and utilize it. The associated docker image includes an interactive jupyter notebook environment, which auto-starts the tutorials. These tutorials are also included into the documentation and accompany the descriptions. In classroom situations, each student can easily start with the interactive tutorials, while the teacher can follow the documentation. The student should implement the core functionality of SciKit-GStat themselves to fully understand geostatistical analysis workflows. This knowledge can then be applied to SciKit-GStat emphasising the correct application of the package and geostatistics in general. Finally, the student can easily apply the learned techniques to real problems with a production-ready Python package. The overall aim is to rather teach geostatistics with the given resources at the example of SciKit-GStat, than narrowing geostatistics down to the application of SciKit-GStat only.

### 5.2 Contributions

Contributions to SciKit-GStat are managed via Github. Generally, anyone can create a private copy of the full source code. Adaptions, enhancements or corrections to the source code of SciKit-GStat can be merged into the official code base via Github. With respect to coding style, technical correctness and overall objective of the library any possible contribution is reviewed by the author or any other maintainer of the package. To further guarantee technical correctness, SciKit-GStat is covered by unit-tests, which test all main functionality in isolated test cases. Due to technical challenges, most plotting routines are not covered by unit-tests. Historically, there have been a number of tests, but they require a lot of maintenance and are to a specific degree dependent on the host platform. Thus, it can be doubted that this is actually beneficial for the user. Additionally, a few tests in the style of end-to-end test (e2e) were added to run a full analysis against an expected result. Such e2e tests also assess the performance, measured as test run-time. However, dropping performance does not cause a test failure, but can be





used by the author and contributors to assess contributions with respect to their influence on performance. It was also decided to not accept any new contributions that decrease test coverage significantly, by adding automatic coverage reports to new

contributions. This can be considered important to assure a specific level of technical correctness for SciKit-GStat, especially because the open source MIT license does not put any warranties in place, that the user could rely on.

### 5.3 Integration into other libraries

The main interface to `gstools` is already discussed in section 4.2. SciKit-GStat has an interface to `pykrige`, which makes it possible to export a `Variogram` instance as kriging parameters directly into `pykrige`. However, as `pykrige` is fun-

damentally changing, it is not yet clear if the interface will still work in the future. Nevertheless, as the code restructuring is finished, the more powerful interface to `gstools` can be used to interact with `pykrige` in a more feature rich, natural and native way.

`scikit-learn` is the most popular data science and machine learning framework in Python. Beside that, `scikit-learn` developed a tool-chain pipeline over the past years, that is used way beyond data science. This enables the user to quickly

change isolated parts of large and complex automated analysis workflows. SciKit-GStat implements an interface to the corresponding class in `scikit-learn`, which makes variogram analysis available in any workflow. At the same time, `scikit-learn` implements a great number of data transformation algorithms as usually used in machine learning. By adopting the pipeline tool-chain, these preprocessing steps can be used together with SciKit-GStat, as many of them are useful for geostatistical preprocessing as well. A prime example is trend detection and detrending, which is often necessary in geostatistics.

## 6 Discussion

Most limitation and notes on application have already been mentioned in the respective sections, along with implementation details. This section is discussing general comments to SciKit-GStat. SciKit-GStat is toolbox for variogram estimation, equipped with a large amount of methods. Most of these methods and settings do not make sense in every situation. SciKit-GStat is generally leaving any assessment of estimated variograms, beyond numerical goodness of fit values, to the user. From this, it

is further clarified, that SciKit-GStat is a variogram estimation toolbox, which is used for building geostatistical methods or conducting analyses. It is not a analysis framework itself.

This limitation also applies to preprocessing. While geostatistical prerequisites, like the intrinsic hypothesis, are mentioned and further literature is referenced, SciKit-GStat does not contain any diagnostic tool to i.e. check given input data any further than by offering the presented scatter plots in figure 3 for visual inspection. External software needs to be used to test and transform

input data. This applies to coordinate transformations as well as observation normalization if required. For both cases flexible and powerful Python packages are available (`scipy`, `numpy`, `scikit-learn`). Hence, I had the impression that anything implemented into SciKit-GStat can't come close to existing software. Furthermore, I cannot claim to overlook all geoscientific fields in enough detail to be able to offer generic integrity checks and preprocessing for just any kind of input data. On the other hand, from my personal experience in answering Github issues, non-transformed, misused and non-applicable datasets





in combination with rather uncommon variogram estimations already lead to some confusion. As an example: If one uses the stable entropy method to find lag classes, the method tries to assure that all classes are of comparable entropy. As a consequence, using the entropy as a variogram estimator will yield nugget effect models by design. If not, it is due to a weakness in method and not a statistical feature of the sample. SciKit-GStat will not stop you from doing so, nor does it stop the user from using this model for external drift kriging, which will solely use the external drift variable for interpolation, then. One might be

under the impression that a sophisticated geostatistical interpolation was performed and the result is backed by the covariance of observations. In fact, one did only apply a computationally intensive averaging overlayed by a simple linear regression of the external drift term. It is up to the user to inspect the Variogram and be aware of these implications. Not everything SciKit-GStat calculates is automatically correct beyond technical correctness.

Another general comment concerns spatio-temporal geostatistics. I want to clearly state here, that spatio-temporal vari-

ograms cannot be exported to any other Python package and SciKit-GStat does not include spatio-temporal kriging. An implementation is neither planned by the author, nor for `gstools` or `pykrige`, as far as I am aware. Thus, from what I can say, one has to use the wonderful `gstat` package and the R programming language, or `gslib` in FORTRAN right now. Due to the lack of kriging procedures, the spatio-temporal variogram representation of SciKit-GStat falls way behind the base class in terms of functionality and interactivity. Similar statements can be made for the directional variogram. While it is as functional,

interactive and powerful as the base class, it can't be exported either. The original intention was to build a diagnostic variography tool for detecting anisotropy. It turned out, that the current design of the directional variogram is incompatible to the design in `gstools` and `pykrige`. Hence, the user has to detect anisotropy and in the case of geometric anisotropy and then transform the input data manually. This can be cumbersome and `gstools` might offer the better approach here, if kriging or field generation are the final steps.

**7 Conclusions**

With SciKit-GStat, the scientific Python community has gained a flexible, well documented and well written package for variogram estimation. SciKit-GStat enables the user to estimate variograms in almost limitless variations in a language-natural and efficient manner. Many quality measures and especially plotting routines accompany the library, to not only *do the hard work*, but also help the user to understand what was actually done. Such an educational aspect of SciKit-GStat is as important as

the technical implementation details. Even the best code can be applied the wrong way to draw incorrect or skewed conclusions. If one does not write the code himself, this risk might be even higher. With SciKit-GStat the focus is on the Variogram. Variograms that are better understood by a user, lead to better models, which are beneficial not only in application, but also as an educational tool.



*Code availability.* The source code of SciKit-GStat is available on Github. Additionally, each minor version is published as a code publi-
cation (Mälicke et al., 2021). The code to reproduce the figures made with SciKit-GStat, including the data samples shown, is available on
Github and Zenodo (Mälicke, 2021). Note that the data samples are also part of the SciKit-GStat documentation.

## Appendix A: Pancake Data

Using a photograph of a pancake for geostatistics was fun, but not only a joke. When I first saw the browning-pattern in the
pan I was just curious if means of geostatistics work for this example as well. The application was easy and straightforward
and I took literally the first photograph made. I find it striking how well the variogram estimation worked. I have no other
geoscientific real world or even artificial data example at hand that yielded more textbook-like variograms than this pancake.
Today, I would conclude that while a pancake is not a geoscientific phenomenon, the browning of the dough is largely driven
by thermodynamic principles which are universally applicable. Thus, this *'artificial'* data set was great for development and
has become my prime benchmark data set for geostatistical method development. I personally prefer artificial datasets over real
world examples here, as sampling sizes and locations can be altered. With real world datasets I, personally, tend to focus too
much on the system that the data actually represents and not the method development. On the other hand, generating a random
field by *putting* a covariance structure represented by a specific variogram into the field and then reproducing the very same
variogram from a sample of the field is not much of a surprise. In these use cases I found pancakes to be very useful.

To bake your own data, there are a few technical instructions, which should help to produce comparable pancakes. The
photograph was taken with a Canon Powershot 540SX digital camera at 3267x2305 resolution. The camera position was as
orthogonal as possible at about 60cm height. The original image was re-scaled to 709x500 pixels by cubic interpolation and
finally cropped to 500x500 pixels centered along the x-axis. To sample the pancake, 300 random pixel positions are chosen,
without replacement to form the array of coordinates. The red band value at these pixels form the corresponding observations
array. The photograph is a PNG, thus the value range is of a unsigned 8-bit integer ($0 <= value <= 255$).
Finally, my pancake dough is very liquid (more like a Crêpe and less like an American pancake). From my experience, liquid
dough and high temperatures (short time in the pan) are the key to spatially structured pancakes. I would expect a classic
American pancake to be way more homogeneous browned. I use 500g of flour, 2 medium sized eggs, about a half liter of milk,
a bit of salt and about 50g sugar. Finally I add water to the dough until it is about as liquid as warm motor oil. Usually, that
sums up at least to another half liter (of water). Maybe a bit more. To bake the random field, use oil, not butter. I could produce
similar results with two different pans on two different stoves (a very old one and a new induction stove). My final advice is to
archive only a digital copy of the pancake and eat the actual one with maple syrup.

*Author contributions.* The manuscript was written by MM. The documentation, user guide and tutorials were written by MM. The source
code of SciKit-GStat was written by MM, with a few exceptions: The directional variogram contains some adaptions by Egil Möller. This
affects the calculation of the direction matrix. The gstools interfaces were in parts written and reviewed by Sebastian Müller. In the early



days of SciKit-GStat, Helge Schneider contributed unit-tests and refactored some functions. Based on Github contributions, more than 99% of the code are authored by MM, in terms of line adaptions.

*Competing interests.* The author declares to have no competing interests.

*Acknowledgements.* First I want to acknowledge the contributors to SciKit-GStat: Egil Möller, Helge Schneider and Sebastian Müller for their valuable contributions. I thank Jon Sheppard for proofreading the manuscript and Erwin Zehe for his valuable feedback on general
structure, content and data usage. I also want to emphasize the indirect contributions by Sebastian Müller, the lead developer of gstools. We spend many hours on discussing the future of Python's geostatistical libraries, which directly affected pykrige, gstools and SciKit-GStat. Thanks to Sebastian, gstools and SciKit-GStat today complement, instead of compete, each other. Finally I thank all the people asking questions and reporting bugs on Github. They made SciKit-GStat a better library and gave me the confidence and the drive to push forward with development, although the paper I wrote the code for in the first place was finished years ago.





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
