# Peer review of "SciKit-GStat 1.0: A SciPy flavoured geostatistical variogram estimation toolbox written in Python."

_Geoscientific Model Development, 2021_

## Author Comment (AC2)

**test_unbinned_fit**

October 19, 2021

**0.1 Test unbinned fit**

```python
[1]: import skgstat as skg
     import numpy as np
     import matplotlib.pyplot as plt
     from scipy.linalg import inv, det
     from scipy.spatial.distance import squareform
     import warnings
     from time import time
     warnings.filterwarnings('ignore')
```

```python
[2]: # use the same dataset as used in GMD paper
     c, v = skg.data.pancake(N=300, seed=42).get('sample')
```

```python
[9]: t1 = time()
     V = skg.Variogram(c,v, bin_func='scott', maxlag=0.7)
     t2 = time() # get time for full analysis, including fit
     print(f"Processing time: {round((t2 - t1) * 1000)} ms")
     print(V)
     fig = V.plot()
```

```
Processing time: 14 ms
spherical Variogram
* * *
Estimator:         matheron
Effective Range:   326.72
Sill:              1584.49
Nugget:            0.00
```

[Figure]

```
[4]: fig = V.distance_difference_plot(show=False)
```

[Figure]

**0.2 ML**

Implementing eq. 14 from Lark (2000), with spherical model, adapted to fit eq. 16 from same publication.

```python
[5]: def f(h, a):
         if h >= a:
             return 1
         elif h == 0:
             return 0
         return (3*h) / (2*a) - 0.5 * (h / a)**3

     def get_A(r, s, b, dists):
         a = np.array([f(d, r) for d in dists])
         A = squareform((s / (s + b)) * (1 - a))
         np.fill_diagonal(A, 1)

         return A

     def like(r, s, b, z, dists):
```

```
    A = get_A(r, s, b, dists)
    n = len(A)
    A_inv = inv(A)
    ones = np.ones((n, 1))
    z = z.reshape(n, -1)
    m = inv(ones.T @ A_inv @ ones) @ (ones.T @ A_inv @ z)
    b = np.log((z - m).T @ A_inv @ (z - m))
**print(b)**
    d = np.log(det(A))
**print(d)**
    loglike = (n / 2)*np.log(2*np.pi) + (n / 2) - (n / 2)* np.log(n) + 0.5* d +␣
 ↪(n / 2) * b
    return loglike.flatten()[0]
```

[6]:
```
from scipy.optimize import minimize

z = V.values#.reshape(len(V.values), -1)
dists = V.distance
fun = lambda x, *args: like(x[0], x[1], x[2], z=z, dists=dists)
t3 = time()
res = minimize(fun, [np.mean(V.distance), np.var(V.values), 0.1 * np.var(V.
 ↪values)], bounds=[[0, V.bins[-1]], [0, 2500], [0, 2400]])
t4 = time()
print(f"Processing time {np.round(t4 - t3, 2)} seconds")
```

Processing time 2.14 seconds

[7]:
```
print('p0: ', [np.mean(V.distance), np.var(V.values), 0.1 * np.var(V.values)])
print('res:', res.x)
```

```
p0:  [253.54190639540656, 1298.8712333333333, 129.88712333333334]
res: [ 184.23866253 1312.89640482    7.23090537]
```

[8]:
```
import matplotlib.pyplot as plt
mod = lambda h: f(h, res.x[0]) * res.x[1] + res.x[2]

x = np.linspace(0, 450, 100)
y = list(map(mod, x))
y2 = V.fitted_model(x)

plt.plot(V.bins, V.experimental, '.b', label='experimental')
plt.plot(x, y, '-g', label='ML fit (Lark, 2000)')
plt.plot(x, y2, '-b', label='SciKit-GStat default fit')
plt.legend(loc='lower right')
plt.gcf().savefig('compare.pdf', dpi=300)
```

[ ]:

---

## Author Response (AR1)

Please find the three *Referee's comments (in italic)* below, followed by my answer and a point-by-point summary of adaptions made to the manuscript, if applicable. The Referee comments are followed by a summary of unrelated additional adaptions made to the manuscript.

**Referee comment #1**

**Specific comments:**

1. T*he pancake dataset was used as example data for the demonstration of the functions in SciKit-GStat and the author described the advantage of it in appendix A. But I still think a complicated real-world data example such as precipitation should be given to support the powerful of the package. Also, such case can give more realistic variogram usage clues to the users.*

**Answer:** As detailed in the discussion, there is already a geoscientific example in use and following the various Referee comments, I revised the manuscript to describe it properly. I also want to clarify, that the pancake is actually a real-world dataset and in my personal opinion it is not a simple dataset. It shows different spatial correlation types at different length scales and is highly sensitve to sampling sizes.

However, I completely rewrote the tutorials of SciKit-GStat and used the Meuse dataset from the R package sp (Pebesma and Bivand 2005, Bivand et al. 2008), for the getting started guide. The main geoscientific workflow (Variogram and Kriging) and the corresponding plottings routines are summarized in the appendix. A detailed scientific geostatistical analysis of this and the already embedded dataset  is from my point of view out of scope for this manuscript, but makes a research paper on its own. Note that the example is far from being a exhaustive, scientific sound analysis of lead contamination.

Adaptions made to the manscript to highlight the used geoscientific dataset and include the new one are in specific:

- Added the Appendix A1 to illustrate a typical workflow using the Meuse dataset. (L. 953-973)

- Added a reference to this appendix into the data description section starting at:  „The meuse dataset is used in the tutorials of SciKit-GStat  [...]" (L. 173-175)

- A whole section about the SoilNet distributed soil temperature time series from the Rott headwater catchment in Fendt was added to the manuscript (L.176-182). The description is largely taken from the associated data description paper (Fersch et al. 2020).

2. *The author declared that the SciKit-GStat version is 1.0, but only version 0.6 (or 0.6.6) of it could be found in github and online document website.*

**Answer:** The referee is right, I should have made this clearer somehow. Version 0.6.X actually is the release candidate for version 1.0. There are no missing versions or something. I wanted to publish the 1.0 version along with changes made to the software due to the discussion. In fact, there were a number of developments in SciKit-GStat, which are related, but also unrelated to this discussion. The most important changes since version 0.6.6 are summarized below:

- Version 0.6.7 introduced a sub-class of MetricSpace that can sample a 2D field in concentric rings to create more uniform distance matrices. This development does not affect the manuscript.

- Version 0.6.8, 0.6.9 & Version 0.6.11 included only bug-fixes for cached calculations, Pyhton version compability and internal NaN value handling.

- Version 0.6.10 changed the K-Means based binning procedure, which is also introduced in the manuscript. In case the K-Means does not converge, an exception is raised.
  Added the describing sentence „Additionally, the K-Means might not converge. In these cases the Variogram class raises an exception and invalidates the variogram." (L. 513-514)

- Version 0.6.12 further enhanced the sample data submodule, to make the tutorials way more straightforward.

- Version 0.6.14 implemented improvements to the plotting routines related to the suggestions of referee #3.

- Finally version 1.0 introduced a utility suite to aid the user in implementing maximum likelihood approaches using SciKit-GStat, which was motivated by the Referee #1 specific comment 4.

3. *Why non-Gaussian geostatistics are not covered in SciKit-GStat?*
**Answer:** non-Gaussian geostatistics are not covered, because I never used them, nor did any of the users ever request non-Gaussian methods. Additionally, I do not oversee the existing literature enough to only summarize, which methods actually all belong to 'non-Gaussian geostatistics'. I basically read the two publications about Copulas (Bárdossy 2006, Bárdossy and Li, 2008) and the generalized sub-Gaussian Model (Guadagnini et al., 2018). In both cases, the workflow is largely incompatible with SciKit-GStat. At the example of Guadagnini et al. (2018): Consider their figure 5 showing the workflow: I am not even sure at which step the non-Gaussian variant of SciKit-GStat would be involved, as some are clearly pre-processing and post-processing (such as process simulations for ie. flow or transport).
To wrap this up: Implementing only what I have heard of non-Gaussian geostatistics so far is already way out of scope for the software as well as the presented manuscript and beyond that, I doubt that the mentioned publications summarize the field of non-Gaussian geostatistics in its entirety.

4. *Why the procedures that can fit a model directly based on unbinned data are not implemented in SciKit-GStat?*
**Answer:** I refer to my exhaustive answer from the open discussion here, which described why an impementation of these methods is largely incompatible with the presented software. However, as suggested by my second anwer (https://doi.org/10.5194/gmd-2021-174-AC2), an additional tutorial has been added to SciKit-GStat with version 0.6.13. The tutorial illustrates how to minimize the implemented theoretical models using a Variogram instance. The tutorial also illustrates, that a custom implementation, without SciKit-GStat, is quite straightforward.
I personally still prefer not to use SciKit-GStat here. The binning into lag classes is deeply built into the variogram class and any analysis code that mixes this class with unbinned approaches seems unintuitve to me. Thus, I implemented the approach used in the tutorial as a utility function. This

makes the approach available to users, but not as a feature of the variogram, which makes more sense to me, from a didactical point of view. Nevertheless the following adaptions were made to the manuscript:

- Appendix C (L. 999-1016) briflly summarizing the new tutorial was added.

- A describing sentence referencing the appendix C is added to the introduction: starting with „Additionally, even a utility suite is implemented, that can build a maximum likelihood function at runtime for any represented variogram to fit a model without binning the data at all (Lark, 2000). Appendix C briefly summarizes the tutorial about maximum likelihood fitting." (L. 79 – L. 81)

**Minor comments:**

1. *Color bars should be plotted in figure 1, 4, and 8.*
**Answer:**

- A new subplot was added to fig. 1, containing the red band of the RGB image, which was used in the manuscript. (P. 6)

- Colorbars were added to fig. 4 (P. 14)

- figure 8 does not containy any continous information, but 10 does. The referee might have confused these two figures. A colorbar has been added to fig. 10 (P. 31)

2. *Line 99, change "SciKit-Gstat" to "SciKit-GStat".*
**Answer:** Changed. (L.101)

3. Line 212, remove one "all".
**Answer:** Changed „all all" to „of all" (L. 223)

4. *Figure 3 was not explained clearly.*
**Answer:**

- The caption of fig. 3 (P. 12) was completely revised and should make sense now.

- The (old)  sentence starting „Along the x-axis [...]"  was removed and replaced with the description starting: „The two subplots show ..." (L. 298-299).

5. *Figure 6, it's better to set transparent color to the surface part so the distribution of the scatter data could be clearer.*
**Answer:** Fig. 6 (P. 16) was updated to show some opacity on the surface. The aspect of the 3D canvas was also changed a little bit. Note that the distribution of the scatter data is still hard to see, which is described as a downside of 3D plots in the manscript (L. 376-379) and motivated the addition of fig. 7.

6. *The data source of figure 6 was not described.*
**Answer:** I refer to the changes summarized in the answer to specific comment 1. These include a description of the data source.

**Referee comment #2**

1. *There are a few writing errors as highlighted in other comments. The explanation of some of the figures is not clear (Fig 3 and 4). Additionally, the accompanying text should also be improved in clarity.*
**Answer:** I carefully proof-read the whole manuscript and corrected writing errors. In relation to fig.3 and 4 the following changes have been conducted:

- The caption of fig. 3 (p. 12) was completely revised and should make sense now.

- The caption of fig. 4 (p. 14) was extended. Note that the figure was adapted following the suggestions of referee #4 (comment 4)

2. *In section 3.4, the default 3D plot and contour plot of Spatio-temporal variograms are presented. However, the text does not explain the data used for this analysis. Simply mentioning the source paper is not enough.*
**Answer:** I refer to the changes summarized in the answer to specific comment 1. of Referee comment #1. These include a description of the data source.

3. *I suggest the authors add variogram analysis' for more complex datasets that better represent the real-world geostatistic analysis as opposed to a simple dataset like pancake.*

**Answer:** I refer to the changes summarized in the answer to specific comment 1. of Referee comment #1.

4. *From my understanding, SciKit-GStat comes with four model fitting algorithms (aside from the approach where the user sets the hyperparameters). However, the author compares the four different distance-based weighting functions only with the Trust-Region Reflective fitting procedure. Was there a reason for choosing this specific procedure for the comparison?*
**Answer:** I wanted to illustrate the differences solely due to the weighting functions and thus kept the fitting procedure fixed. Trust-Region Reflective was used, as it is the default option in SciKit-GStat.

5. *Additionally, why is there no comparison presented between the different fitting procedures?*

**Answer:** For all variograms presented throughout the manuscirpt as well as in the tutorials the different fitting procedures find almost the same parameters, which makes a visual comparison difficult.

**Referee comment #3**

1. *A grid should be embedded in Figure 1 to make visualisation easier.*
**Answer:** A grid was added to fig 1 (P. 6)

2. *A histogram can be plotted separately along with Figure 2 with fitted normal distribution so that it's easier to visualise data distribution and accuracy of prediction.*
**Answer:** While SciKit-GStat allows making separated plots, I personally prefer the way it is

plotted. The histogram is visualizing the count of point pairs for each of the distance lags, on top of the corresponding lag class. Which I find quite useful. Unfortunately, I am not sure what I should fit a normal distribution to and what ,prediction' is referring to, here.

However, the figure caption of fig 1 (P. 6) was extended to a description of the histogram.

3. *A grid should be embedded in Figure 4 with more distinguishable colour bands to simplify visualisation.*
**Answer:**

- A grid was added to all subplots of fig. 4 (P. 14)

- The colorband was changed to the same as used in fig. 10.

4. *How did you plot figure 6? Which data is used? A paragraph on data description can be provided*
**Answer:** I refer to the changes summarized in the answer to specific comment 1. of Referee comment #1. These include a description of the data source.

5. *In figure 7, what does the legend show? a more descriptive legend should be provided.*
**Answer:**

- The label was added to fig. 7 (P. 17)

- The caption of fig. 7 (P. 17) was updated.

- Additionally, since version 0.6.14 SciKit-GStat is now always showing the label by default

6. *In figure 9, the axis should be labelled.*
**Answer:** Labels were added to the axes of fig. 9 (P. 30); Additionally, the legend font size was increased.

7. *Figure 10 can be redrawn with high resolution along with an axis labelled*
**Answer:**

- Fig. 10 was redrawn with maximum possible resolution. Note that the underyling field is represented by a 100x100 matrix, which can't be increased. Thus, one can see pixels on the figure.

- A colorbar was added to both rows of subfigures. All subplots share the same value range

- The figure caption was updated.

- An axis label was added

8. *Make figure 11 a bit bigger*
**Answer:**

- Fig. 11 was changed from a one-column to a two-column figure (12 instead of 8.3 cm)

- The axes were labled

**Additional adaptions made to the manuscript:**

There are a few additional adaptions made to the manuscript:

- The directional variograms shown in fig. 5 (P. 15) were calculated using (old) wrong parameters. Now, the variograms as described in the manuscript are shown. The figure was updated.

- Changed „in situ" to „in-site" (L. 26)

- Changed „many current" to „other" (L. 39)

- Removed „has" (L. 45)

- Replaced „scipy" with „SciPy" (L. 117)

- Added „a" (L. 138)

- Changed „„ close to geostatistical textbooks, where the variogram is always the first geostatistical method introduced" to „close to geostatistical textbooks, which usually present the variogram first." (L. 138 – 139)

- Corrected the wrongly referenced R-package ‚gstat' to the R-package ‚sp' (L. 162 – L. 163)

- Replaced „modules" with „packages" (L. 187)

- added „parameter" on L. 214

- replaced „be generalizable to the data set." with „be generalized to the random field." (L. 277)

- An explaining sentence was added to the description of fitting methods (4.1.5) to avoid misunderstandings . „ith version 1.0 maximum likelihood and a likelihood function is used in two different parts of SciKit-GStat, but actually doing two very different things (only the optimization part is the same…). Added: „Note that this approach is optimizing the variogram parameters by their likelihood of fitting to the experimental data, it is not a maximum likelihood fitting of the variogram model to the sample auto-corrleation as described i.e. by Lark (2000). The latter approach is briefly described in appendix C. (L. 705 – L. 707)

- Changed „Kriging" to „kriging" on L.  262; 364; 649; 827; 828; 830; 833; 839

- Changed „Variogram" to „variogram" on L. 784; 832; 928; 947

- The reference to gstools was updated (Müller and Schüler 2021, to Müller et al, 2021). The work was referenced on L. 101; 855

- The reference to SciPy was updated (Virtanen et al. 2020)

**References**

Bárdossy, András. "Copula-based geostatistical models for groundwater quality parameters." Water Resources Research 42.11 (2006).

Bárdossy, András, and Jing Li. "Geostatistical interpolation using copulas." Water Resources Research 44.7 (2008).

Guadagnini, Alberto, Monica Riva, and Shlomo P. Neuman. "Recent advances in scalable non-Gaussian geostatistics: The generalized sub-Gaussian model." Journal of hydrology 562 (2018): 685-691.

Lark, R. M. "Estimating variograms of soil properties by the method-of-moments and maximum likelihood." European Journal of Soil Science 51.4 (2000): 717-728.

Marchant, B. P., and R. M. Lark. "Robust estimation of the variogram by residual maximum likelihood." Geoderma 140.1-2 (2007): 62-72.

Pebesma EJ, Bivand RS (2005). "Classes and methods for spatial data in R." R News, 5(2), 9–13. https://CRAN.R-project.org/doc/Rnews/.

Bivand RS, Pebesma E, Gomez-Rubio V (2008). Applied spatial data analysis with R, Second edition. Springer, NY. https://asdar-book.org/.

Fersch, Benjamin, et al. "A dense network of cosmic-ray neutron sensors for soil moisture observation in a highly instrumented pre-Alpine headwater catchment in Germany." Earth System Science Data 12.3 (2020): 2289-2309.